# Functional control of a 0.5 MDa TET aminopeptidase by a flexible loop revealed by MAS NMR

Diego F. Gauto[1,8], Pavel Macek[1,9], Duccio Malinverni[2], Hugo Fraga[1,3,4], Matteo Paloni[5], Iva Sučec[1], Audrey Hessel[1], Juan Pablo Bustamante[6], Alessandro Barducci[5✉] & Paul Schanda[1,7✉]

Large oligomeric enzymes control a myriad of cellular processes, from protein synthesis and degradation to metabolism. The 0.5 MDa large TET2 aminopeptidase, a prototypical protease important for cellular homeostasis, degrades peptides within a ca. 60 Å wide tetrahedral chamber with four lateral openings. The mechanisms of substrate trafficking and processing remain debated. Here, we integrate magic-angle spinning (MAS) NMR, mutagenesis, co-evolution analysis and molecular dynamics simulations and reveal that a loop in the catalytic chamber is a key element for enzymatic function. The loop is able to stabilize ligands in the active site and may additionally have a direct role in activating the catalytic water molecule whereby a conserved histidine plays a key role. Our data provide a strong case for the functional importance of highly dynamic - and often overlooked - parts of an enzyme, and the potential of MAS NMR to investigate their dynamics at atomic resolution.

[1] Univ. Grenoble Alpes, CEA, CNRS, Institut de Biologie Structurale (IBS), 71, Avenue des Martyrs, F-38044 Grenoble, France. [2] Department of Structural Biology and Center for Data Driven Discovery, St Jude Children's Research Hospital, Memphis, TN, USA. [3] Departamento de Biomedicina, Faculdade de Medicina da Universidade do Porto, Porto, Portugal. [4] i3S, Instituto de Investigacao e Inovacao em Saude, Universidade do Porto, Porto, Portugal. [5] CBS (Centre de Biologie Structurale), Univ Montpellier, CNRS, INSERM, Montpellier, France. [6] Instituto de Bioingenieria y Bioinformatica, IBB (CONICET-UNER), Oro Verde, Entre Rios, Argentina. [7] Institute of Science and Technology Austria, Am Campus 1, A-3400 Klosterneuburg, Austria. [8] Present address: ICSN, CNRS UPR2301, Univ. Paris-Saclay, Gif-sur-Yvette, France. [9] Present address: Celonic AG, Eulerstrasse 55, 4051 Basel, Switzerland. ✉email: barducci@cbs.cnrs.fr; paul.schanda@ista.ac.at

Cells use large protein assemblies to perform many essential biological processes. The cellular protein quality control machinery comprises a collection of such large protein assemblies, including chaperones, unfoldases, proteases and peptidases. Collectively, these proteins eliminate damaged or misfolded proteins either by refolding them to a functional state or by proteolysis. Many proteases and peptidases form large oligomeric assemblies, often in the molecular weight range of hundreds of kilodaltons. The self-compartmentalization of these machineries allows for specificity, as only unfolded proteins or small fragments can access the protease reaction centers. The proteasome, a prominent example, cleaves proteins to peptides of ca. 7–15 residues length[1]. These peptide fragments are then further digested to amino acids by aminopeptidases[2], such as tetrahedral aminopeptidases, present in all forms of life. While structures of many proteases, peptidases, and chaperones are available, the precise mechanisms of their action, including substrate entry, fixation, and product release, often remain difficult to decipher. Motions and allosteric regulation are intimately linked to enzymatic function, as shown for machines of the protein quality-control system[3–5]. An increasing number of cases reveals that enzymatic turnover can directly depend on the inter-conversions of states, such as conformations in which the active site is open or closed[6–10] or where larger domains reorganize e.g., for binding additional accessory proteins[11]. Characterizing the link between enzyme structure, dynamics, and function at the atomic scale remains, however, experimentally challenging.

We study here the 468 kDa large tetrahedral aminopeptidase TET2 from the hyperthermophilic archaeon *Pyrococcus horikoshii*, a member of the metallo-peptidase family M42. Archaeal TET aminopeptidases and homologous structures in other organisms[12,13] assemble to dodecameric tetrahedral structures, encapsulating twelve $Zn_2$ active sites within a large hollow lumen with a diameter of ca. 60 Å[14–17] (Fig. 1). Pores on each of the four faces of the tetrahedron, each ca. 18 Å wide, allow the passage of unfolded or short α-helical, or β-hairpin peptides while preventing folded proteins from entry to the catalytic chamber. The processing occurs in a sequential manner from the N-terminus of the peptide[18]. The fastest hydrolysis is observed for peptides of up to ca. 12 amino acids length, and the longest peptides processed by TET2 are ca. 35 residues long[14]. The active center contains two metal ions, labeled M1 and M2 in the nomenclature of Schechter and Berger[19], separated by circa 3.5 Å. Besides the catalytic role of the two metals, they are important for assembly to dodecamers[20,21]. In particular, site M1 is implicated in the stabilization of the oligomeric interface. The two metal sites are generally occupied by zinc. Site M1, but not M2, can be exchanged to $Co^{2+}$. The M2 site is considered as the catalytic metal that hosts the catalytically active water molecule, and sits in the specificity pocket that hosts the side chain of the substrate. A third metal site, M3, has been found recently, adjacent to the active site, and shown to broaden the substrate specificity[20]. The binuclear active site is chelated by histidines (H323 at site M1; H68 at M2) and carboxylates (E213, E212, D235, D182; the latter bridges both zinc ions), and a water molecule[15].

The proposed catalytic mechanism is common to binuclear metallo-aminopeptidases, such as Leucyl-aminopeptidase (LAP) and AAP[12,22,23]. In this mechanism, peptide bond cleavage proceeds via the activation (deprotonation) of a water molecule that bridges the zinc ions. The water molecule donates a proton to a conserved glutamate (E212 in TET2), and the remaining hydroxyl group then attacks the peptide bond of the substrate[15,22,24]. The tetrahedral intermediate state, a diol, where one oxygen stems from the attacking OH and one from the carbonyl group, is stabilized by interaction with the zinc ions. Insight into this state comes from structures obtained with inhibitors; in particular, the

transition-state substrate analog amastatin binds with affinity in the nanomolar range[22] with its scissile bond positioned at the binuclear metal center (Supplementary Figure 1). The proton that was donated by the water molecule and resides on Glu212 is then transferred to what becomes the new N-terminus of the peptide, thus completing the bond cleavage. The cleaved amino acids may be released from the TET particle either through small pores located on the tetrahedral faces close to the central entry pore[15], or through small pores at the apices of the tetrahedron[18], or the large entry pores. The kinetics of the reaction proceeds with a rate constant, $k_{cat}$, of the order of up to 50 s$^{-1}$ at ambient temperature and several thousand s$^{-1}$ at the physiological temperatures of the hyperthermophilic TET enzymes[20]. TET can also process substrates when it is in its dimeric form (which is also found in vivo) but the catalytic efficiency is strongly reduced compared to the dodecameric assembly, in particular towards large substrates. Thus, the compartmentalization to dodecameric hollow assemblies has an important role for activity[18].

Different TET isoforms have different substrate preferences and may assemble into hetero-dodecameric assemblies with improved efficiency for peptide processing: mixed dodecameric assemblies comprising subunits of TET1, TET2 and TET3 may be tuned to degrade entire peptides that may remain within the catalytic chamber throughout the sequential degradation[16,18,25,26]. Homo-dodecameric TET2 displays highest activity for cleavage of

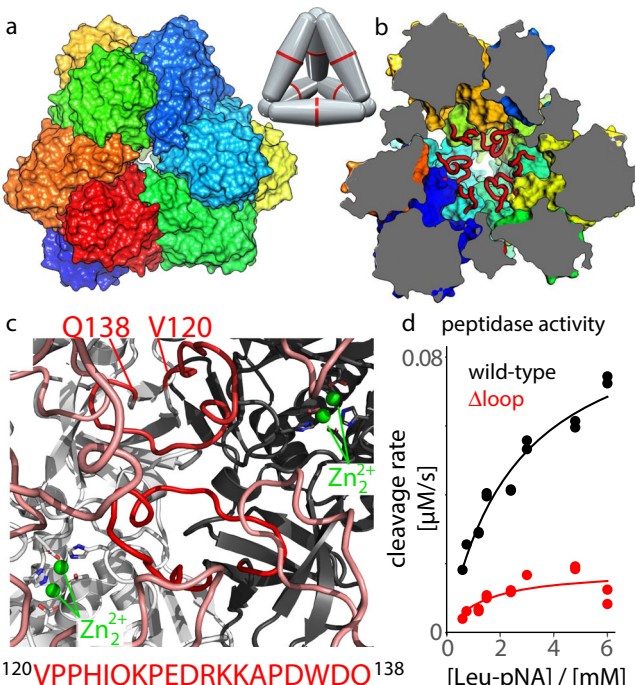

**Fig. 1 Structure and activity of the TET2 aminopeptidase.** Assembly of the dodecameric tetrahedron seen from outside (**a**), with one of the four entry pores in the center, and in a cut-open view (**b**), where the loops are shown in red. Different subunits are shown with different colors. The structure is based on PDB ID 1Y0R, and the loops (unresolved in the crystal structure) have been modeled with Swiss-Model. The schematic model shows the arrangements of the six dimers within the dodecamer. **c** View into the catalytic chamber and onto two adjacent subunits (light grey, black) with their respective loops (red), and the catalytic zinc ions in the active sites, as well as loops from adjacent subunits. **d** Enzyme kinetics data, obtained as the initial rate of absorbance signal following the cleavage of leucine-p-nitroanilide (Leu-pNA). Data points at identical substrate concentration indicate duplicate measurements. Source data are provided as a Source Data file.

hydrophobic residues, with a preference for leucine as the N-terminal amino acid, showing a high activity up to 100 °C over a broad pH range[27]. Eukaryotic homologs of TET2 are involved in blood pressure regulation in humans[28,29] and hemoglobin degradation in the human malaria parasite *P. falciparum*[30].

Although static high-resolution structures are available, important mechanistic aspects of the function of TET peptidases remain debated, including substrate entry and trafficking within the lumen of the TET particle, as well as the release of amino acids[18]. To understand such mechanistic details, characterizing the dynamics and interactions within the protein and with substrates is of great interest. Solution-state NMR is ideally suited to study protein dynamics and thus link structures to function. However, for proteins of the size of TET2, solution-NMR suffers from rapid signal loss for most sites. Methyl-specific labeling is often the only way to obtain site-specific information for proteins beyond ca. 100 kDa[31–34]. This approach is by definition limited to methyl-bearing residues.

Unlike solution-state NMR, magic-angle spinning (MAS) NMR does not face inherent protein size limitations and allows to see, in principle, each atom. MAS NMR is a powerful technique for studying dynamics at atomic resolution, and has been applied to sedimented, crystalline, membrane and amyloid proteins[35–47]. Most of the previous MAS NMR dynamics studies focused on proteins below 20–30 kDa, as the resonance overlap often encountered in larger proteins complicates analyses. We have recently achieved the resonance assignment of ca. 90% of the backbone atoms, and about 70 % of the side-chain heavy atoms, as well as of methyl groups of Ile, Leu and Val residues[48,49] and Phe ring C-H moieties[50] in TET2. With 353 residues per subunit, TET2 is among the largest proteins for which such near-complete assignment has been achieved. This assignment, along with distance restraints, has allowed us to develop an approach that uses medium-resolution cryo-EM data along with (primarily solid-state) NMR data to solve the structure of TET2[48].

We use here quantitative MAS NMR experiments, co-evolution analyses and molecular dynamics (MD) simulations to probe at the atomic level the dynamic contacts formed between the active sites and a functionally important loop. Enzyme kinetics experiments and mutants allow linking these findings to the function of this enzyme. Our study provides direct insight into the functional control of an enzyme through a region which is not even visible in high-resolution crystal structures, and demonstrates the maturity of MAS NMR for studying the structure-function link of even very large proteins.

## Results

**A highly dynamic loop in the catalytic chamber controls enzyme activity**. The catalytic chamber of TET2 comprises twelve long loops, one from each subunit. Interestingly, in 3D structures of TET2 obtained by crystallography[15], these loop regions have not been modeled (residues 120–132 are missing); similarly, in our recent cryo-EM data[48] this region had very weak electron density. In crystal structures of the homologous TET1 and TET3, the loop has been modeled partially (5 and 9 residues are missing in PDB 2wyr and 2wzn, respectively); the modeled residues have B-factors well above average (see Supplementary Figure 1, which also lists the loop residues that have not been modeled). In MD simulations we confirm that these loop regions in TET1 and TET3 are dynamic, even though they have been partially modeled (discussed further below). In a homologous peptidase that forms the same dodecameric assembly, such as those from *Streptococcus pneumoniae* (PDB 3kl9) a 15-residue long stretch is missing, similarly to TET2. All these observations point to large mobility of these loops. When modelled into the TET2 structure, the loops

fill close to 30% of the catalytic chamber volume (Fig. 1a–c). Thus, one may assume that they represent a significant steric penalty for substrate trafficking in the catalytic chamber, raising the question of their possible functional role.

To probe whether the loops influence catalytic activity, we measured the peptidase activity of a TET2 mutant in which the loop has been shortened to a two-residue $\beta$-turn ($\Delta$(120-138), henceforth called $\Delta$loop mutant). Cleavage of the peptide bond of a small substrate, leucine-p-nitroanilide (H-Leu-pNA), is detected by the absorbance of the reaction product, pNA. The $\Delta$loop variant showed a dramatically reduced enzymatic activity compared to the wild-type protein (Fig. 1d and Supplementary Table 1). To ensure that this loop shortening does not significantly impact the protein structure which would lead to the observed drop in activity, we have collected MAS NMR correlation experiments (2D hNH and 3D hCONH). Based on the observation of very similar chemical shifts of the $\Delta$ loop and WT variants (discussed below, Fig. 3) we can rule out structural distortions in this mutant, and it must be the loop itself which plays an important role for catalytic activity of WT TET2.

We used MAS NMR to probe the conformational behavior of the loop in more detail. In dipolar-coupling based MAS NMR experiments, which are inherently most sensitive for rigid parts, the backbone of residues S119 to K132 and W136 to Q138 could not be assigned[48]. The absence of these peaks in such dipolar-coupling transfers may point to large-amplitude motions that would render the dipolar-coupling based transfer inefficient. In the presence of very fast motion (tens of nanoseconds at most), scalar-coupling based transfers shall lead to efficient transfer. We collected such scalar-coupling based hNH correlation experiments, but we did not observe additional peaks (Supplemantary Figure 2).

The $^{15}N$ backbone amide site of the assigned and resolved residue D135 shows rapid $^{15}N$ $R_{1\rho}$ spin relaxation ($\geq 12\,s^{-1}$; Supplementary Figure 3a), pointing to motions in the nanosecond-to-millisecond range. We have performed additional $^{15}N$ $R_{1\rho}$ Bloch-McConnell relaxation-dispersion experiments[51]. The relaxation-dispersion profile of D135 is non-flat, which shows that this residue undergoes microsecond dynamics (Supplementary Figure 3c). Collectively, the absence of observable signals for most of the backbone sites in the loop, and the direct evidence from spin relaxation of D135 indicates that the loop undergoes $\mu$s motion.

In an additional experiment that aims to detect the loop signals we exploited a specific and very sensitive isotope-labeling method of phenylalanines which we developed recently[50]. In this isotope-labeling scheme, a single $^1H$-$^{13}C$ pair in the para-position of the Phe ring is introduced in an otherwise fully deuterated sample. We have shown previously that it allows the sensitive detection of the side chain of Phe. We have prepared such a sample in which we positioned a Phe at position 123 of the loop; for reasons outlined in detail further below this position was chosen. We, thus, expected to see an additional correlation peak in the $^1H$-$^{13}C$ spectrum of the phenylalanine (para-CH) labeled sample of H123F TET2. However, the experiment only features the ten native Phe sites, but not the F123 (Supplementary Figure 4). This observation, although indirect, is another hint to the presence of large-scale $\mu$s-ms dynamics of the loop.

To gain further insight into the loop dynamics, we used methyl-directed MAS NMR of a specifically Ile/Leu/Val $^{13}CHD_2$-labeled and otherwise deuterated sample (Fig. 2a). The well-resolved cross-peaks of Val 120, located toward the beginning of the loop, and of Ile 139, located at the C-terminal junction of the loop to a $\beta$-strand, were assigned through a mutagenesis approach[49]. The additional methyl group signal in the loop, $\delta 1$ of Ile 124, is not spectrally resolved[49]. V120 and I139 are convenient probes of the loop conformational dynamics, which we quantitatively measured.

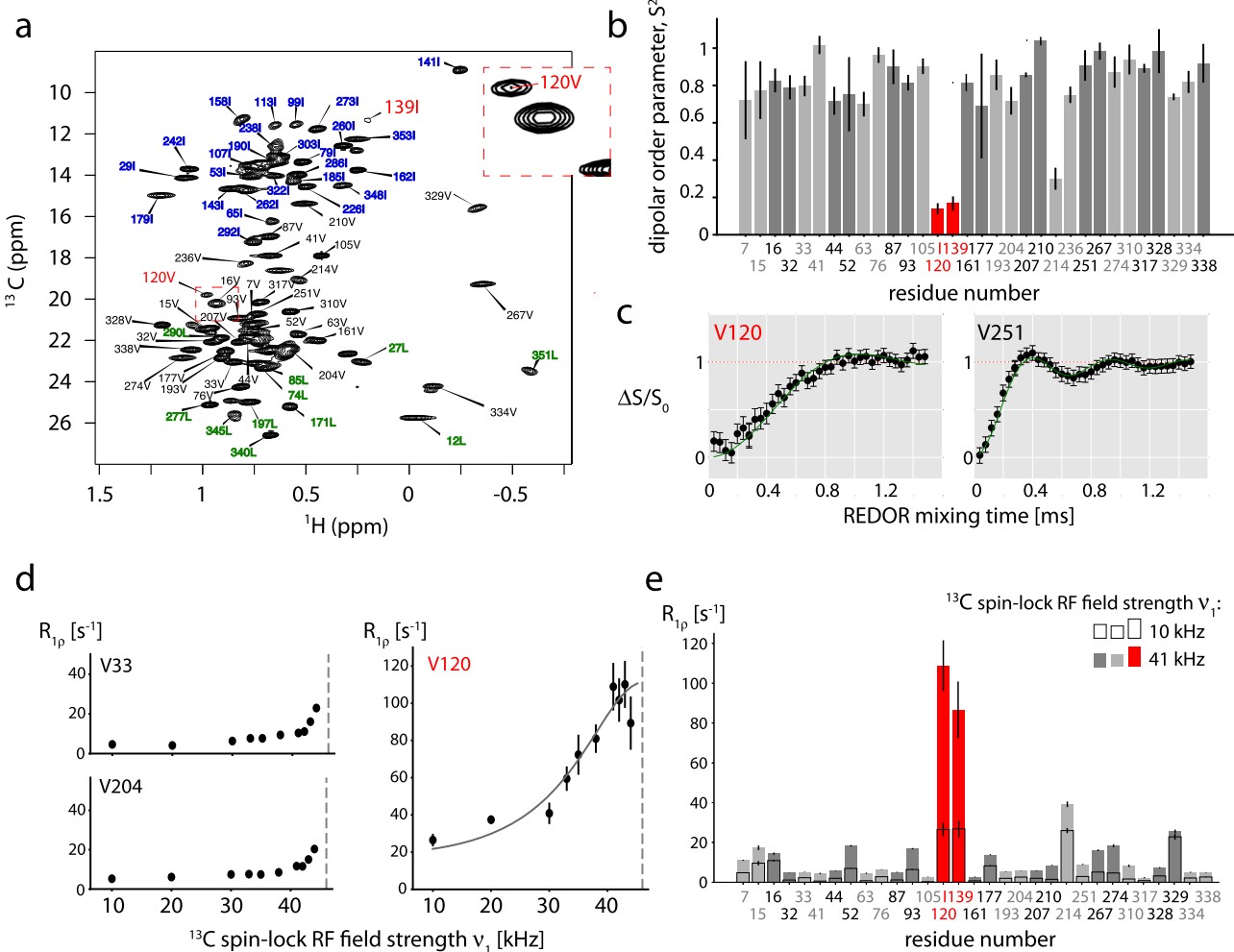

**Fig. 2 The loop samples a wide conformational space within the cavity. a** Methyl spectrum of u-[$^2$H,$^{15}$N],Ile-$\delta$ 1-[$^{13}$CHD$_2$],Val-$\gamma$2-[$^{13}$CHD$_2$],Leu-$\delta$ 2-[$^{13}$CHD$_2$] labeled TET2. **b** Dipolar-coupling-derived $^1$H-$^{13}$C order parameters of all Val-$\gamma$2 CHD$_2$ methyl groups and one Ile-$\delta$ 1 CHD$_2$ (Ile139), derived from a Rotational Echo Dipolar Recoupling (REDOR) experiment at 55.555 kHz MAS frequency. Data of all Val-$\gamma$2, Ile-$\delta$1, and Leu-$\delta$2 methyls are shown in Supplementary Figure 5. Data for the two sites in the loop are highlighted in red. **c** Example REDOR curves. All REDOR curves are shown in Supplementary Figure 6. **d** Example $^{13}$C R$_{1\rho}$ profiles, up to the NERRD regime, where the $^{13}$C spin-lock radio-frequency (RF) field strength approaches the MAS frequency (46 kHz, dashed vertical line). Data for all methyl groups are reported in Supplementary Figure 7. See Supplementary Figure 9 for discussion of the estimated time scale. **e** $^{13}$C R$_{1\rho}$ values at two $^{13}$C spin-lock field strengths (i.e., two points of the NERRD profile), highlighting that only V120 and I139 have a strong NERRD effect. Data in panels (**b**) and (**e**) are presented as best-fit values (in terms of minimal chi-square) $+/-$ one standard deviation, where the standard deviation has been determined by Monte Carlo error estimation, based on three times the spectral noise level (see Methods). Source data ($^{13}$C relaxation data, REDOR data and analysis scripts) are provided as a Source Data file.

Fig. 2b, c shows the $^1$H-$^{13}$C dipolar-coupling tensor data of all Ile-$\delta$1, Val-$\gamma$2, Leu-$\delta$2 methyl groups. The motion-averaged dipolar-coupling tensor reflects the motional amplitude of the methyl group axis, averaged over all time scales up to hundreds of $\mu$s[52], and can be directly translated to the order parameter, which ranges from 1 for fully rigid to 0 for fully flexible sites. In the case of a valine, it corresponds to motion of a single sidechain torsion angle ($\chi$1) and of the backbone. While the vast majority of valines are rather rigid, with order parameters ($S^2$) in the range 0.7 to 1, Val120 is highly flexible ($S^2 = 0.15 \pm 0.02$). Ile 139 also displays a similarly low order parameter. It is noteworthy that the two probes may underestimate the full loop mobility: Val 120 is adjacent to a residue that still had been modeled in the crystal structure, although with high B-factor (Ser 119); Ile 139 is located in a 4-residue short helix adjacent to the loop (again with high B-factors). Thus, while the two probes clearly have the largest amplitude among all methyl-bearing residues we observed, with order parameters below ca. 0.2, the actual loop motion may be even larger.

Spin-relaxation rate constants are sensitive to both the time scales and the amplitudes of dynamics. So-called near-rotary-resonance relaxation dispersion (NERRD) experiments are particularly informative of $\mu$s-ms motion[53]. In this type of experiment, the relaxation rate constant in the presence of a spin-lock pulse, R$_{1\rho}$, is measured at different spin-lock radio-frequency (RF) field strengths up the the regime where the RF field nutation frequency reaches the MAS frequency $\nu$, also termed the $n=1$ rotary-resonance condition ($\nu_{RF,13C} = \nu_{MAS}$)[51,54,55]. The profile of R$_{1\rho}$ as a function of RF field strength strongly increases as the RF field strength approaches the MAS frequency only if the bond undergoes $\mu$s motion. Unlike in Bloch-McConnell relaxation-dispersion experiments[56] often used in solution-state NMR, NERRD experiments sense $\mu$s-ms dynamics even if the exchanging conformations do not differ in their chemical shifts. Figs. 2d, e show NERRD data for Val sites in TET2. The NERRD curves of almost all methyl sites are flat or show only a modest increase in R$_{1\rho}$ ($\leq$10 s$^{-1}$) close to the rotary-resonance condition (see

Supplementary Figure 7). Strong non-flat NERRD profiles are observed for V120 and I139 and unambiguously demonstrate that these sites undergo $\mu$s motions. As the motion experienced by the V120 side chain is presumably complex, involving methyl rotation, side chain motion and loop reorientation, quantitative analysis is challenging, and we limit the analysis to an estimated time scale of ca. 10–1000 $\mu$s (see Supplementary Figure 9 for discussion). Motion on this time scale leads to fast transverse relaxation in MAS NMR experiments[52], which provides an explanation for the broadening beyond detection of most backbone signals and the elevated $R_{1\rho}$ and relaxation dispersion of D135 (Supplementary Figure 3).

We additionally measured longitudinal relaxation rate constants ($^{13}$C $R_1$), which are sensitive to faster motions (nanoseconds)[52,57,58]. Neither V120 nor I139 have particularly fast $R_1$ decay, demonstrating that their motion does not take place on the ns time scale (Supplementary Figure 8).

We then characterized which parts of the catalytic chamber are in (possibly transient) contact with the loop. We exploited the fact that the chemical shift is a suitable reporter of such contacts: it is sensitive to the local environment around a given atom, averaged over all conformations sampled on time scales up to milliseconds according to their relative population. Therefore, even transient contacts of a given residue with the loop would be imprinted on the chemical shifts of its atoms (as long as the corresponding conformations have a sizeable population level). It shall, thus be possible to detect which residues are in contact with the loop by comparing the chemical shifts of a wild-type (WT) protein and a protein lacking the loop. We have prepared these two samples of TET2 (WT, $\Delta$ loop) with uniform labeling (u-$^2$H,$^{15}$N,$^{13}$C, in H$_2$O buffer) and probed the backbone $^1$H$^N$, $^{15}$N, and $^{13}$C$\alpha$ chemical shifts. Figure 3a–c shows the hCANH-derived chemical-shift differences, between WT and $\Delta$ loop, which reflect the effects induced by the loop. As expected, the CSP effects are located in the interior of the enzymatic lumen. The large area within the catalytic chamber involved in loop contacts spans residues from the entry pore to the active site. Transient contacts of the loop with all these residues requires large-amplitude motion, in line with the dynamics data reported in Fig. 2.

**Co-evolution and molecular dynamics simulations detect loop contact sites.** Based on the observation that the loop is crucial for function (Fig. 1d) and in contact with many residues, we reasoned that interaction patterns of the loop may be conserved across TET homologs. Thus, we investigated how residues in the loop co-evolved in more than 20,000 different homologous sequences. Analysis of co-evolution (Direct Coupling Analysis, see Methods) highlights the conservation of contacts involving the loop and residues in the catalytic chamber, both intra- and inter-molecularly (Fig. 3d). Co-evolutionary couplings arise from statistical correlation in a multiple-sequence alignment (MSA). Such co-evolution in a loop region is remarkable, as generally the residues in loop regions evolve quickly[59]. Co-evolution is observed between residues of the loop and e.g., V93, located right next to the Zn$_2$ center, and P246 in the entry pore. It is noteworthy that the couplings can be explained only assuming a certain degree of flexibility of the loop, because a static loop is unable to fulfill the co-evolved contacts. Our analysis, therefore, implicitly suggests that the motion itself is conserved by evolution and functionally relevant in the family (or in a significant fraction of it).

We used one-microsecond-long all-atom molecular dynamics (MD) simulations of the dodecameric TET2 assembly to gain additional insight into the contacts of the loop with other structural parts. These simulations are challenging for several

reasons: with its 468 kDa, TET2 represents a size challenge for all-atom MD, making it difficult to study long time scales; furthermore, as the experimental data revealed, the loop motion occurs on a tens-of-microseconds time scale (Fig. 2 and Supplementary Figure 9). Consequently, in order to obtain convergence from MD simulations, hundreds of microseconds to milliseconds would need to be simulated. Our simulations can, thus, only provide qualitative conclusions, and these are in very good agreement with the experimental observations. Figure 3e highlights the residues within the TET2 cavity which are in transient contact along the MD trajectory; these span the range from the entry pore to the active site, mirroring the NMR CSP data and the co-evolution data. The MD data also allow identifying numerous contacts between the loops of two adjacent monomers within the dimeric building block of TET2, as well as contacts to loops from other subunits (Supplementary Figure 11).

MD provides the possibility to obtain a structural view of the loop conformations. In particular, we aimed to understand which role the evolutionarily-conserved contacts play for the loop conformations. We observed that the loop conformations in which the loop forms contacts to residues D94 and I238 (P121-I238 and P122-D94 and Q125-D94) correspond to states which bring the loop in proximity to the active site (Fig. 3f). This finding suggests that the observed co-evolution may be related to contacts of the loop to substrates. Collectively, three fundamentally different approaches, MAS NMR, MD, and co-evolution analysis, reveal large amplitude motion of the functionally important loop. The MAS NMR relaxation-dispersion data show that this process occurs on a time scale of ca. 10 $\mu$s to 1 ms. Interestingly, simulations of TET1, TET2, and TET3 dimeric assemblies also show that this loop is a very dynamic structural element, suggesting that at least within the archaeal TET assemblies the loop flexibility is retained, although the PPH motif (discussed below) is less dynamic in TET1 and TET3 than in TET2 (Supplementary Figure 12).

**Conserved loop residues are important for enzyme-substrate interaction.** To identify the mechanisms of this enzymatic control via a highly flexible loop we investigated the sequence alignment of TET2 homologs, and find a strong conservation of a histidine in this loop, and to lower extent also of a Pro-Pro motif, corresponding to P121, P122 and H123 in TET2 (Fig. 4a). Analysis of the structures of related aminopeptidases and homologs in which the loop has been modeled into the electron density suggests that H123 of a given subunit may be in close vicinity to the active site of the adjacent subunit (Supplementary Figure 1). However, among the crystal structures there is a remarkable variability of the distances between the His and the active site, ranging from ca. 4 Å to over 20 Å. In the cryo-EM structure of TET2, i.e., the sample studied here, the His is over 20 Å away from the active site (Supplementary Figure 13). Histidines often play an important role in enzyme catalysis because the imidazole side chain allows it to form hydrogen bonds and to combine donor and acceptor properties[60]. We envisioned two possible manners how the His in the loop (which is not part of the active site) may play a role in the enzymatic process. On the one hand, the His may stabilize the substrate in the active site. Indeed, hydrogen bonds formed between the substrate and a residue outside the canonical Zn$_2$ center play such a stabilizing role in several other aminopeptidases[22]. We speculated that the highly conserved histidine H123 may contribute to stabilizing the substrate in the active site; the Pro-Pro motif preceding H123, is conformationally restricted[61], and may be important to position the conserved His within the active site. Such a substrate-stabilizing effect shall be reflected in a lowered

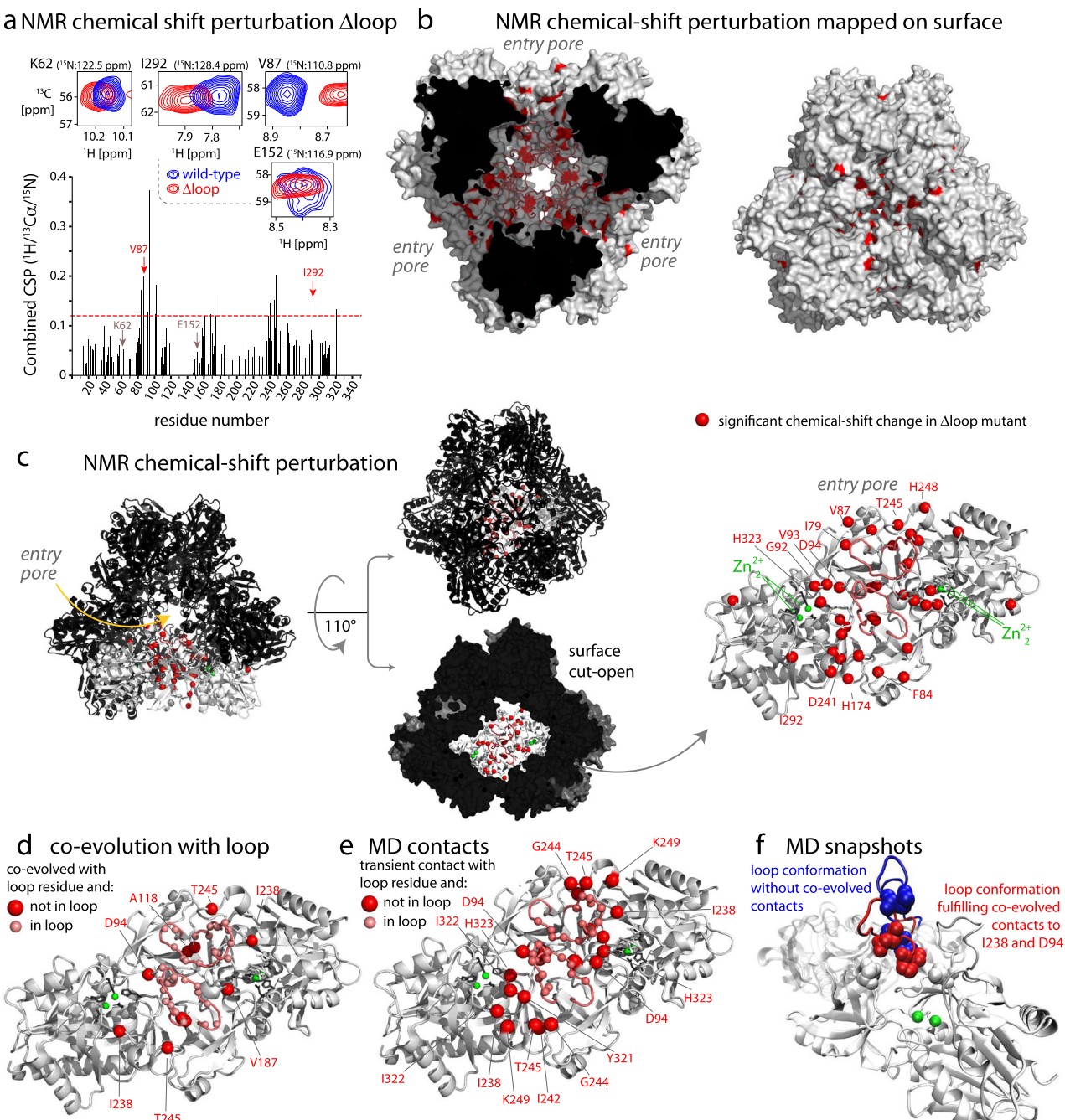

**Fig. 3 The conformational space sampled by the loop detected by MAS NMR, MD and co-evolution analysis. a** Backbone $^1$H-$^{13}$Cα-$^{15}$N chemical-shift differences of wild-type and Δ loop TET2 in 3D hCANH experiments, shown by selected $^1$H-$^{13}$Cα planes (top) and residue-wise combined CSP values (see methods). **b**, **c** Plot of significant CSP values (3 times standard deviation, indicated by a horizontal line in **a**) on the structure of TET2. In **c**, a dimer (grey) is highlighted, by first showing it in the context of the dodecameric assembly, then rotating and extracting this dimer (right). **d** Co-evolution analysis, showing residues which co-evolved with residues in the loop (residues 120–138), plotted onto the structure of a TET dimeric subunit in the same orientation as in **c**. Red spheres are residues (outside the loop) which co-evolved with residues from the loop (light red spheres). The co-evolution data are listed in Supplementary Table 2. **e** Plot of residues in transient contact with the loop, as observed during the all-atom MD trajectory. Residues were considered in contact when the minimum inter-residue distance between any heavy atoms was below 5 Å. Residues in the loop that are highlighted with spheres have a transient contact with loop from the adjacent subunit. **f** Snapshots of the MD trajectory, in which the evolutionary contacts of loop residues are formed (red) or absent (blue).

Michaelis constant (compared a mutant without these residues). On the other hand the His may play a more active "chemical" role, such as assisting in the nucleophilic attack by activating a water molecule. If the histidine plays a role in the chemistry of the reaction, then one might be able to see the signatures e.g., in pH-dependent effects.

We investigated the role of the conserved residues using functional assays with mutant proteins. Specifically, we mutated H123 to either phenylalanine or tyrosine (ring structure with similar dimensions as His, but without the H-bond donor/ acceptor nitrogens) or lysine (to investigate the importance of a positive charge). In an additional mutant, Δ(122,126), we

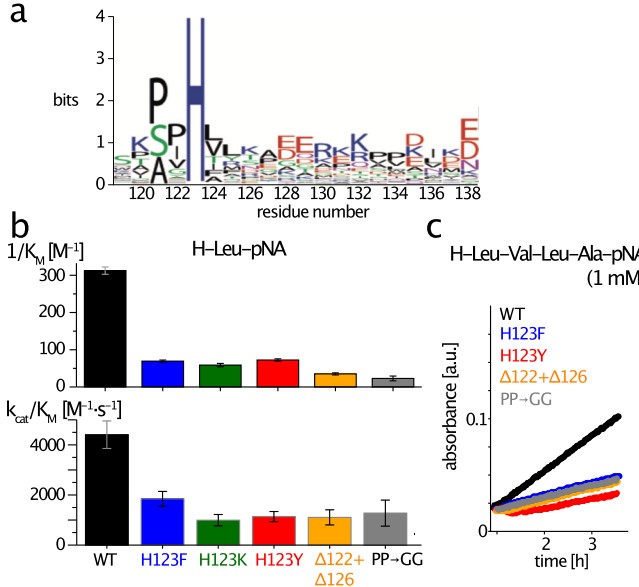

**Fig. 4 Functional importance of H123, the PP motif and the loop length for enzymatic activity. a** Logoplot showing that H123 and the two preceding prolines are highly conserved. **b** Results of a Michaelis-Menten enzyme kinetics assay of TET2 with the short chromogenic substrate H-Leu-pNA. The inverse of the Michaelis constant, $1/K_M$, and $k_{cat}/K_M$ (insert) are shown. The individual points are the best-fit values from individual experiments (replicates). The bar heights and the error bars were obtained from all the replicates. **c** Time traces of enzymatic assays with a longer peptide substrate, H-Leu-Val-Leu-Ala-pNA (1 mM concentration). The limited solubility of the peptide hampered systematic evaluation of $K_M$ and $k_{cat}$, but the results are in good qualitative agreement with those of H-Leu-pNA. Data are presented as best-fit values (in terms of minimal chi-square) $+/-$ SEM, where the SEM has been determined from a Monte-Carlo based approach described in detail in the Methods section. Source data (kinetic time traces of enzyme kinetics) are provided as a Source Data file.

shortened the loop by one residue on each side of the H123, thus reducing its ability to reach into the active site. To test the importance of the PP motif (residues 121 and 122), we replaced it by a flexible GG stretch.

Figure 4b shows the results of activity assays for the WT and mutant samples processing the chromogenic substrates H-Leu-pNA. The mutants have significantly reduced activity. In particular, the Michaelis constant[62], $K_M$, of the mutants indicates that the stability of the enzyme-substrate complex is reduced by up to one order of magnitude (Fig. 4b and Supplementary Table 1). Experiments with a longer substrate, the tetrapeptide H-Leu-Val-Leu-Ala-pNA, are in good qualitative agreement with the data from the short H-Leu-pNA (Fig. 4c). Taken together, the activity assays show that the stability of the enzyme-substrate complex is reduced through mutations that render the loop either shorter or remove a residue of the conserved Pro-Pro-His motif.

**Ligand-dependent conformational equilibrium of the loop.** How can a highly dynamic loop without a stably defined position play a crucial role for the activity of an enzyme? We propose that within the wide range of loop conformations there are states which bring important residues, such as H123, close to the active site; in these conformations, contacts to the substrate may increase the stability of the enzyme-substrate complex, as evidenced by the $K_M$ values. According to this view, the equilibrium of loop conformations is expected to be altered by the presence of substrate bound to the active site.

We experimentally tested this model by measuring the effect of bound ligands on the conformational equilibrium of the loop. The challenge for such experiments comes from the short life time of substrates inside the active site, as they are cleaved within milliseconds[20]. Moreover, the population of TET2 particles that have simultaneously all 12 sites occupied is extremely small. As an experimentally feasible alternative to generate a temporally stable and fully ligand-occupied state of TET2, we prepared samples of TET2 with the inhibitor amastatin (Fig. 5a), a peptide that tightly and non-covalently binds to the active sites of TET2[15]. MAS NMR experiments reveal the chemical-shift perturbations induced by this tightly bound inhibitor (Supplementary Figures 14 and 15a). CSPs are observed in the close vicinity of the binding site, in excellent agreement with the crystal structure[15]. Importantly, a single set of resonances is found in our samples, i.e., the entire population is shifted from the apo state to the amastatin bound one.

We turned to methyl $^1$H-$^{13}$C correlation spectra and used the signal of V120 to monitor whether the bound inhibitor impacts the conformational equilibrium of the loop. As compared to the apo state, the cross-peak of V120 shifts significantly upon amastatin binding (Fig. 5c). V120 is ca. 6–18 Å away from the nearest atom of the inhibitor in the MD ensemble (Fig. 5b), too far to cause any impact of amastatin on the V120 signal by direct molecular contact. Because the chemical shift reports on the ensemble-averaged conformational equilibrium, the altered peak position of V120 rather reveals that the relative population levels of the loop conformers are altered; similarly, the backbone N-H signal of D135 is significantly altered upon amastatin binding (Supplementary Figure 15a). Figure 5e sketches this idea of a conformational ensemble.

Based on the activity measurements, we expected that H123, via its effect on stabilizing the substrate in the active site, plays an important role in reshuffling the loop conformational equilibrium upon ligand binding. Consequently, we expected that a H123F mutant, unable to form these contacts, would be unable to induce this population re-shuffling. We found that this is exactly the case: in H123F TET2, the reporter NMR resonance of the loop conformation, V120, was essentially unaffected by the presence of the inhibitor in the active site (Fig. 5d). We ensured that amastatin tightly binds to the H123F mutant, evidenced by significant CSPs in $^1$H-$^{15}$N correlation spectra upon inhibitor binding and again a fully bound state (i.e., no residual apo-state peaks), akin to the wild-type protein (Supplementary Figure 15b).

**Binding and release at the active site depend on the loop.** Having shown that the presence of ligand in the active site alters the loop conformation, we investigated if the reverse is equally true, i.e., if the presence of the loop alters binding and release of a substrate (or a non-cleavable weak binder) in the active site. Because substrates are rapidly degraded by TET2, it is difficult to study their binding and release under equilibrium conditions. We discovered by serendipity that the dialcohol 2-methyl-2,4-penta-nediol (MPD), a common crystallization agent used also to obtain solid-state NMR samples, interacts with the active site of TET2. The inhibitory properties of aliphatic alcohols on aminopeptidases have been reported earlier[63]. In the presence of MPD, several H-N moieties (Gly92 (NH), Asp94 (NH), and the zinc-chelating histidine 323 (Nδ-Hδ)) feature two cross-peaks of approximately equal intensity, indicating that MPD-bound and free states of TET2 co-exist in slow exchange (Fig. 6). A comparison with spectra of TET2 without MPD (sedimented rather than MPD-precipitated) shows that one of these two peaks corresponds to an MPD-bound form.

Whereas in the WT TET2 we find, thus, two peaks in the presence of MPD (MPD-bound and free), we only find a single

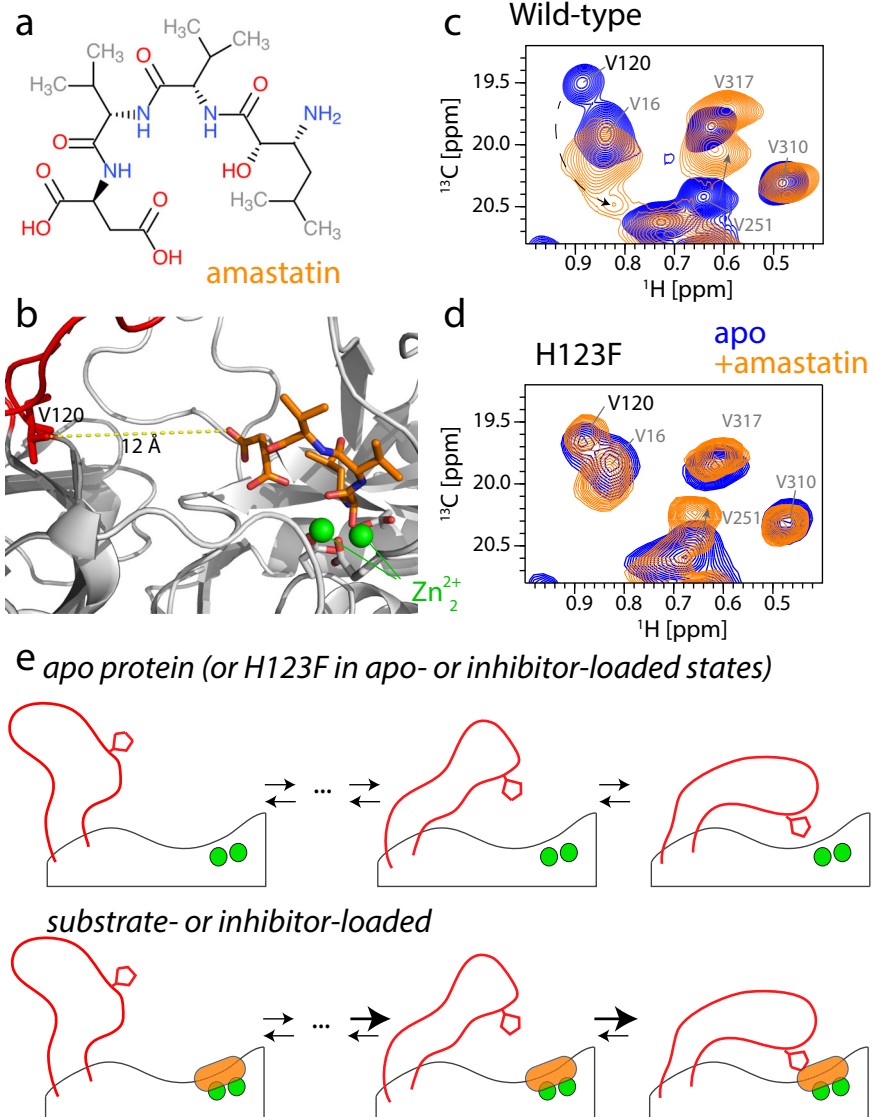

**Fig. 5 The loop conformational ensemble is altered by ligand-binding to the active site. a** Chemical structure of the inhibitor peptide amastatin.
**b** Structural view of amastatin in the active site (PDB: 1Y0Y; loops modeled with SwissModel). **c** $^1$H-$^{13}$C correlation spectrum of WT TET2 in the absence and presence of the inhibitor amastatin, showing a clear change of the peak of the loop residue V120. The dashed arrow indicates the putative shift of the V120 cross-peak to a new position. The shift of the cross-peak of V251 is ascribed to effects from direct binding of amastatin. **d** Equivalent $^1$H-$^{13}$C spectra of H123F TET2 in the apo and amastatin-bound states. The unchanged peak position and intensity of V120 indicates that the loop ensemble is not altered by amastatin, despite full occupancy of the active site with amastatin (Supplementary Figure 15). The full spectra are shown in Supplementary Figure 16. **e** Schematic representation of the loop conformational ensemble, and the impact of ligands in the active site on reweighting populations within the ensemble.

peak in the Δ loop mutant (Fig. 6). For all three sites, G92, D94, and H323 (Nδ-Hδ), the observed peak is close to the peak position that, in WT TET2, corresponds to the free (not MPD-bound) state. This finding suggests that the MPD-bound state is not significantly populated in the absence of the loop. Of note, these experiments do not provide direct evidence that the loop directly interacts with the ligand (in this case MPD). It is conceivable that the loop stabilizes the MPD-bound state more indirectly, by contacting other residues of the protein rather than the substrate itself. The precise mechanism, as well as the binding affinity, likely depends also on the nature of the substrate. Irrespective, this data shows that the observed affinity of a ligand at the active site directly depends on the presence of the loop, mirroring the reduced binding affinity of substrates that we observed in the activity measurements (Fig. 4, 1/$K_M$ values).

We questioned whether the conserved His may also play a role for the catalytic reaction itself. We first investigated whether addition of free histidine might suffice to increase the activity in a His-free mutant (H123Y). The answer is no: addition of free His not only does not increase the activity, but abolishes the function of WT and mutant TET2, possibly because of binding of the free amino acid to the active site (Supplementary Figure 17). We then performed activity measurements with WT and mutants (H123F, H123Y, H123K, Δ(122,126)) at higher pH values (pH 9.3). Given the fact that the mechanism involves nucleophilic attack by a hydroxide ion, increasing the pH is expected to lead to a significant acceleration. Indeed, the mutants that lack the H123 have higher activity at the higher pH of 9.3. (The increase is modest, which might be due to negative effects on the structural integrity.) However, for WT TET2, the activity is essentially

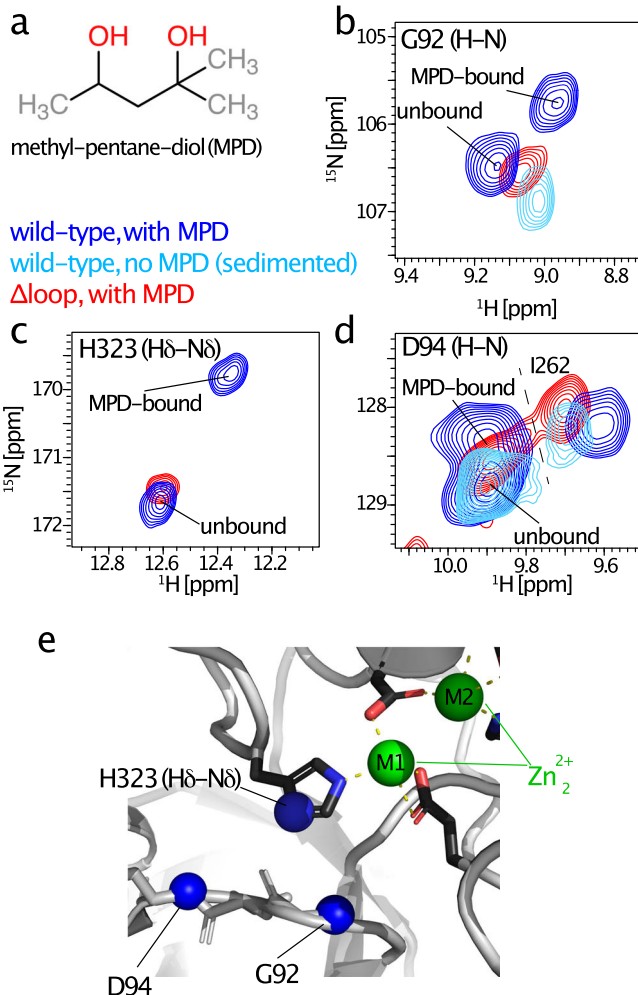

**Fig. 6 Binding of a weak inhibitor at the active site is altered in the Δloop mutant. a** Chemical structure of 2-methyl-2,4-pentanediol (MPD), a previously[63] identified competitive inhibitor for aminopeptidases. **b–d** Zoom on the $^1$H-$^{15}$N correlation peaks of the backbone amides of G92 and D94, and the side-chain Nδ position of H323, one of the chelators of the zinc active site. In the sample obtained by precipitating TET2 in 50% (vol/vol) MPD, two peaks are visible; comparison to the sample of ultracentrifuge-sedimented TET2 (light blue) reveals that one of these is MPD-bound and one unbound. In the MPD-precipitated Δ loop mutant, only one peak is visible, which, based on the similarity of the peak position, we assign to the state not bound to MPD. **e** Location of the three $^{15}$N sites with peak doubling in the presence of MPD.

unaltered compared to pH 7.5 (Supplementary Figure 18). A plausible explanation of these findings is that H123 and its free electron pair on the imidazolium side chain facilitate abstraction of the proton from water (which is thought to occur on E212). This would explain why already at pH 7.5 the reaction proceeds as fast as at pH 9.3. The mutants lack this possibility, and thus are less active at pH 7.5. (Note that over the pH range the singly protonated imidazolium is predominant[64].) Given the rather modest difference in pH-response for WT and mutants, we believe, however, that this role of H123, if present, is minor compared to substrate stabilization.

## Discussion

Why do the large TET aminopeptidases, present in all kingdoms of life, feature loop regions which fill up almost one-third of their catalytic chamber? Why has evolution generated these long stretches which seem to hamper the access of substrates to the active sites, rather than having an empty spacious lumen? Our combined MAS NMR, functional and computational study clarifies the functional role of these loops, which we show to be highly flexible (Fig. 2). We demonstrated that these loops act to stabilize substrates in the active site (higher enzyme-substrate affinity, Fig. 4). The loop–substrate interaction in turn shifts the conformational ensemble of the loop (Fig. 5e), and the loop has an impact on active-site binding of a ligand, i.e., it seems to favor the bound state (Fig. 6). Crystallography (Supplementary Figure 1) and MD simulations (Supplementary Figure 12) point to motion of these loops also in other TET isoforms, although the effect appears most pronounced in TET2.

Residues in loop regions evolve rapidly, on evolutionary time scales, and are generally hardly conserved[59]. Evolution of the physico-chemical characteristics of the loop may have helped to widen the substrate specificity of TET peptidases. Remarkably, though, we identified a highly conserved His within a Pro-Pro-His stretch, and demonstrated its functional relevance. Interestingly, histidines in loop regions close to the substrate have been identified also in the peptidases APP and eMetAP[65,66], and also proposed for thermolysin[67]. Although the structural scaffolds of these peptidases are unrelated to TET2, it has been found in crystal structures with ligands trapped in the active sites that the imidazolium nitrogen of histidine side chains interacts with oxygens of the bound ligand (Supplementary Figure 19). Our data indicate that H123 of TET peptidases may act similarly; the histidine is indeed able to reach the substrate for forming such interactions (Supplementary Figures 13 and 20). A possible additional role of the histidine might be to polarize water molecules, thereby increasing their nucleophilic properties. Given the modest effect (Supplementary Figure 18) we assume that this role is minor.

Why is this functionally important element highly flexible, rather than being located on a short, less flexible element in the direct vicinity of the active site? We propose that the high degree of flexibility is required to allow the passage of substrates within the chamber, particularly as the substrates can be up to 35 amino acids long. Freedom of movement within the chamber is important not only for newly entering substrates but also for substrates that were cleaved once and which remain in the chamber for further degradation at one of the 12 catalytic sites[18]. The length and flexibility of the loops may furthermore allow the required versatility for the interaction with a broad range of substrates of different lengths.

On the methodological side, the current study establishes that MAS NMR is highly suited for probing enzyme function, even of very large complexes such as the half-megadalton large TET assembly. For complexes of this size, solution-state NMR[68,69] is generally limited to methyl groups. The ability of MAS NMR to detect essentially all backbone and side chain sites allows to obtain a more comprehensive view; here, only the combination of methyl data with backbone and even side chain His resonances allowed seeing with parts are in contact with the loop (Fig. 3), or binding the ligands (Fig. 6). We have exploited advanced MAS NMR methods to probe dynamics, including $^{13}$C NERRD data (Fig. 2d), which, to our knowledge is the first report of this method, and asymmetric dipolar-coupling tensor averaging (Fig. 2b), both of which are unavailable for solution-state NMR methods. The prerequisite for performing such studies at the atomic level is that the individual cross-peaks are visible and resolved. In the present case, the spectral resolution is, generally, very high, even at modest magnetic field strength (600 MHz $^1$H Larmor frequency). While similarly high resolution has been reported for a number of other systems by MAS

NMR, the present approach is not necessarily general for any system. Also, due to the extensive $\mu s$ mobility, most of the backbone of the key loop here went undetected. In the general case, a combination of methods such as MAS NMR, solution-NMR and other spectroscopic and structural methods along with simulations may be required, depending on the molecular system. Such insight into conformational dynamics exploration might be decisive to reveal the connections between static structures to functional mechanisms.

## Methods

**Protein samples.** TET2 from *P. horikoshii* (UniProt entry O59196) was produced by overexpression of a pET41c plasmid encoding the TET2 sequence in *Escherichia coli* BL21(DE3) (Novagen) cells in suitably isotope-labeled M9 minimum media (for all NMR) or LB medium (for functional assays). Samples used for NMR studies were either u-[²H,¹³C,¹⁵N] labeled (for all 3D H-N-C correlation experiments, Figs. 3 and 6), or u-[²H,¹⁵N],Ile-δ 1-[¹³CHD₂],Val-γ2-[¹³CHD₂],Leu-δ 2-[¹³CHD₂] labeled (data in Figs. 2c and 5). The sample of amastatin-bound WT TET2 (Fig. 2d) was u-[²H,Val-γ2-[¹³CHD₂],Leu-δ 2-[¹³CHD₂] labeled. The labeling of deuterated samples was achieved by using M9 minimum culture media in 99.8 % D₂O, the use of ¹⁵NH₄Cl as sole nitrogen source, and D-glucose (deuterated and ¹³C₆-labeled for u-[²H,¹³C,¹⁵N] samples or deuterated, not ¹³C labeled, for methyl-labeled samples). Proteins for enzymatic assays were produced in a similar manner (temperature, growth time), but in LB medium.

E. coli BL21(DE3) cells were transformed with the pET41c-PhTET2 plasmid (kanamycin resistance). For production of deuterated samples, the cells were adapted to M9/D₂O medium in three steps (preculture in LB/H₂O during day, preculture in M9/H₂O over night, preculture in M9/50% H₂O/50% D₂O during day, M9/D₂O over night). For the culture, cells were grown to an OD₆₀₀ of ca. 0.6. At this point, for the methyl-labeled samples, ketoacid precursors were added, according to the manufacturer's instruction (NMR-bio, www.nmr-bio.com), and protein expression was induced 1 hour later by isopropyl-β-D-1-thiogalactopyranoside (IPTG, 1 mM in final culture). The culture was grown at 37 °C for 4 h before harvesting by centrifugation.

For protein purification, the cell pellet was resuspended in lysis buffer [50 mM Tris, 150 mM NaCl, 0.1 % Triton X-100, lysozyme (0.25 mg/ml), deoxyribonuclease (0.05 mg/ml), 20 mM MgSO₄, and ribonuclease (0.2 mg/ml) (pH 8)]. Cells were disrupted in a Microfluidizer using three passes at 15,000 psi. The extract was heated to 85 °C for 15 min followed by centrifugation at 17,500 relative centrifugal force (rcf) for 1 hour at 4 °C. The supernatant was dialyzed overnight against 20 mM Tris and 100 mM NaCl (pH 7.5) at room temperature and centrifuged at 17,500 rcf for 10 min at 4 °C. The supernatant was loaded on a Resource Q column (GE Healthcare) and TET2 was eluted with a linear gradient [0 to 1 M NaCl in 20 mM tris (pH 8) over 10 column volumes]. The fractions containing protein with similar mass (39 kDa), according to SDS-polyacrylamide gel electrophoresis (12.5% polyacrylamide), were pooled and concentrated using an Amicon concentrator (Millipore) with a molecular mass cutoff of 30 kDa. The protein solution was then loaded onto a HiLoad 16/600 Superdex 200 column (GE Healthcare) equilibrated with buffer containing 20 mM Tris (pH 8) and 100 mM NaCl.

Samples for MAS NMR measurements were prepared as described earlier[48]; briefly, TET2, was concentrated to 10 mg/mL in 100% H₂O buffer containing 20 mM Tris, 20 mM NaCl (pH 7.6), and mixed 1:1 (vol/vol) with 2-methyl-2,4-pentanediol (MPD), which results in appearance of white precipitate, which we filled into 1.3 mm MAS rotors (Bruker Biospin) using an ultracentrifuge device (ca. 50,000 *g*, in a Beckman SW32 rotor, 20,000 rpm) for at least 1 hour. We have also prepared samples by sedimenting TET2 from the buffer solution [20 mM Tris, 20 mM NaCl (pH 7.6)] with the same ultracentrifuge parameters, over night, without addition of precipitation agent (used for data shown in Fig. 6, light blue). ¹³C-¹³C spectra of MPD-precipitated, isopropanol-precipitated and sedimented samples were highly similar, and also similar to solution-state NMR spectra (Supplementary Figure 21). Note also that the MAS NMR spectra are very similar to solution-state NMR spectra (see Figure S1 of reference [48]).

The loop-deletion mutant plasmid, lacking residues 120–138 of the WT sequence, was prepared by the RoBioMol platform at IBS Grenoble within the Integrated Structural Biology Grenoble (ISBG) facility. The other mutants were generated by a commercial provider, GenScript.

**NMR.** MAS NMR data were acquired on a 14.1 T (600 MHz ¹H Larmor frequency) Bruker Avance III HD spectrometer (Bruker Biospin) using a 1.3 mm probe tuned to ¹H, ¹³C, and ¹⁵N frequencies on the main coil, and an additional ²H coil that allows for deuterium decoupling, which greatly enhances resolution of ¹³CHD₂ spectra[70]. One additional data set, a ¹³C R₁ measurement, was collected on a 22.3 T (950 MHz ¹H Larmor frequency) Bruker Avance III HD spectrometer (Bruker Biospin) using a similar ¹H, ¹³C, ¹⁵N, ²H 1.3 mm probe. The effective sample temperature in all experiments was kept at ca. 28 °C, using the water frequency, $\delta_{H2O}$, as chemical-shift thermometer (and an internal DSS as chemical-shift reference), whereby the temperature $T$ (in °C) is related to the bulk water chemical

shift $\delta_{H2O}$ as $T = 455-90\cdot\delta_{H2O}$. This temperature calibration was found to be in good agreement with an independent temperature calibration via KBr chemical shifts[71] in an external sample; we used the water-based temperature calibration throughout this study.

Three-dimensional hCANH, hCONH and hcaCBcaNH experiments[72] were recorded on u-[²H,¹³C,¹⁵N] labeled wild-type and ΔTET2 at 55 MHz MAS frequency and 600 MHz ¹H Larmor frequency. The experiments used cross-polarization steps (i) from ¹H to ¹³C at typical RF field strengths of ca. 90 kHz (¹H, linear ramp 90–100%) and 35 kHz (¹³C) and a typical duration of 2 ms, (ii) from CA or CO to ¹⁵N at typical RF field strengths of ca. 40 kHz (¹⁵N, linear ramp 90–100%) and 14 kHz (¹³C), typically for ca. 8 ms, and (iii) from ¹⁵N to ¹H at typical RF field strengths of ca. 95 kHz (¹H, linear ramp 90–100%) and 40 kHz (¹⁵N), for ca. 1 ms. The additional CA-CB (out and back) transfer step in the hcaCBcaNH was done with a INEPT transfer, using a 6 ms total transfer delay. The selective pulses for the CA-CB transfer had a REBURP shape (70 ppm bandwidth). The selective pulses for homonuclear decoupling (CO from CA and vice versa) in the indirect ¹³C dimensions were ISNOB (applied to the ¹³C spins to decouple) and REBURP (applied to the ¹³C spins in the transverse plane, for the Bloch-Siegert correction element), akin to previously reported experiments[73]. All experiments are implemented in the NMRlib library[74] and freely available for academic users (http://www.ibs.fr/nmrlib).

The combined ¹H, ¹³Cα, ¹⁵N CSP reported in Fig. 3a was calculated as CSP = $\sqrt{(\Delta\delta(^1H))^2 + \alpha_N(\Delta\delta(^{15}N))^2 + \alpha_{CA}(\Delta\delta(^{13}CA))^2}$, where $\alpha_{CA} = 0.3$ and $\alpha_N = 0.1$, and $\Delta\delta$ denote the chemical-shift differences in the two spectra in units of ppm.

All ¹³C relaxation experiments and the REDOR experiment described below were obtained using pulse sequences reported in Figure S2 of ref. [50] as a series of 2D ¹H-¹³C spectra (also implemented in NMRlib[74]). ¹H-¹³C transfers (out and back) were achieved by cross-polarization, typically using ca. 2 ms long CP transfer with a ¹H RF field strength of ca. 90 kHz (linear ramp 90–100%) and matching the ¹³C RF field strength to the $n = 1$ Hartmann–Hahn condition (i.e., ca. 35 kHz). ¹³C near-rotary-resonance relaxation dispersion (NERRD) $R_{1\rho}$ experiments[51,53] (Fig. 2d, e) were recorded at 14.1 T and a MAS frequency of 46 kHz. Relaxation delays were adapted in the different experiments, in order not to damage the hardware with extensively long high-power spin-lock duration; the delays are listed in Supplementary Table 3.

¹³C R₁ measurements were done at 22.3 T, using relaxation delays of 0.05, 0.2, 0.4, 0.6, 0.8, 1.0, 1.25, 1.5, 2.0, 2.5 s.

¹H-¹³C rotational-echo double resonance (REDOR)[75] experiments (Fig. 2c), in the implementation described in ref. [76] were used to measure asymmetric dipolar coupling tensors. The MAS frequency was 55.555 kHz (18 μs rotor period). The ¹H and ¹³C π pulses were 5 μs and 6 μs (100 kHz and 83.3 kHz RF field strength), respectively. One out of two ¹H π pulses was shifted away from the center of the rotor period, in order to scale down the dipolar-coupling evolution and thus sample it more accurately, as described earlier[77], such that the short and long delays between successive ¹H π pulses were 0.5 μs and 7.5 μs, respectively.

NMR data were processed in the Topspin software (version 3, Bruker Biospin) and analyzed using CCPnmr[78] (version 2.3) and in-house written python analysis routines. In analyses of the NERRD experiment, a two-parameter monoexponential decay function was fitted to the spin-lock-duration-dependent peak intensity decays at the various RF field strengths.

The fitting procedure of the REDOR experiment was described previously[76]. Briefly, numerical simulations were performed with the GAMMA simulation package[79] (version 4.3), setting all pulse-sequence related parameters (MAS frequency, pulse durations, RF field strengths and timing) to the values used in the experiment. A series of such simulations was carried out, in which the ¹H-¹³C dipolar-coupling tensor anisotropy was varied from 1030 to 15,000 Hz (where a rigid H-C pair at a distance of 1.115 Å has a tensor anisotropy of 43,588 Hz, which results in a rigid-limit value of 14,529 Hz when considering the fast methyl rotation) with a grid step size of 30 Hz, and the tensor asymmetry was varied from 0 to 1 with a grid step size of 0.05. Each experimental REDOR curve was compared to this two-dimensional grid of simulations (ca. 9800 simulations in total) and a chi-square value was calculated for each simulation. The reported best-fit tensor parameters are those that minimize the chi-square. Error estimates were obtained by a Monte Carlo approach (pages 104–109 of reference [80]). Briefly, for each methyl site 1000 synthetic noisy REDOR curves were generated around the best-fit simulated REDOR curve, using the spectral noise level and assuming a normal distribution for generating the noisy data points, within three times the standard deviation of the noise level of the spectra. These 1000 synthetic REDOR curves were fitted analogously to the above-described procedure, and the standard deviation over the tensor anisotropy and asymmetry is reported as error estimates. Squared order parameters (Fig. 2b) were obtained by dividing the best-fit tensor anisotropy by the rigid-limit value (14,529 Hz), and squaring the value.

**MD simulations.** All MD simulations were performed with Gromacs 2018.3[81] using amber99sb-disp[82] force field for the protein and TIP4D[83] model for water molecules. The v-rescale[84] and Parrinello-Rahman[85] schemes were employed to control temperature ($T = 300$ K) and pressure ($P = 1$ atm) respectively. A cutoff of 1 nm was used to compute van der Waals interactions, while electrostatic interactions were evaluated by means of the Particle Mesh Ewald algorithm using a

cutoff of 1 nm for the real space interactions. The LINCS[86] algorithm was used to restrain all bond lengths to their equilibrium value. High-frequency bond-angle vibrations of hydrogen atoms were removed by substituting them by virtual sites, allowing an integration time step of 4 fs[87]. Initial configuration for the TET2 dodecameric complex was taken from X-ray structure (pdb code: 1Y0R) and the missing loop was modeled with Swiss Model. The TET2 dodecamer was solvated in a rhombic dodecahedron box with a volume of 2880 nm$^3$ with periodic boundary conditions. Distance restraints between protein molecules and zinc atoms were applied to preserve the local geometry of the enzymatic site. In the simulations of the substrate-bound protein, the substrate was modeled as a tetrapeptide (Leu-Leu-Val-Ala) where the N-terminal residue was modified in order to have a neutral terminus. A substrate molecule was bound to the active site of each monomer by introducing an additional set of distance restraints between substrate and zinc atoms. These restraints were modeled on the basis of the X-ray structure of amastatin-bound complex (pdb code: 1Y0Y) to preserve a correct binding geometry. Apo and substrate-bound systems were energy minimized and equilibrated for 200 ns and then 1 $\mu$s production runs were performed for each system. Reported results were obtained by analyzing one frame every 100 ps. Residues were considered in direct contact when the minimum inter-residue distance between heavy atoms was below 5 Å, whereas a looser cutoff (7 Å) was considered when evaluating DCA predictions according to standard practices in coevolutionary analysis[88]. Initial structures for TET1 and TET3 dimeric assemblies were taken from X-ray structures with pdb codes 2WYR and 2WZN, respectively. Chains A and C were used for the TET1 system, while chains A and D were used for the TET3 complex. Missing residues of both systems were rebuilt with MODELLER using the interface available in UCSF Chimera 1.11[89]. Initial structure for the dimeric assembly of TET2 complex was generated by combining the X-ray structure (pdb code: 1Y0R) with the conformation of the 115–143 fragment obtained by electron microscopy (pdb code: 6R8N). Each dimeric complex was solvated in rhombic dodecahedron boxes with a volume of 1480 nm$^3$ with periodic boundary conditions, and simulated for 1 $\mu$s.

**Enzymatic activity assays**. The enzymatic activity was measured by following the absorbance change induced when a para-nitroanilide (pNA) labeled substrate is enzymatically cleaved using aminoacyl-pNA compounds H-Leu-pNA and H-Leu-Val-Leu-Ala-pNA (Bachem, Bubendorf, Switzerland) as substrates. Measurements were performed on a BioTek Synergy H4 plate reader (Fisher Scientific) measuring the absorbance at 410 nm in a 384-well plate at 50 °C. In all cases, the wells were filled with 50 $\mu$L of substrate solution at concentrations varying in the range from 0.1 to 6.4 mM for H-Leu-pNA and 1 mM for H-Leu-Val-Leu-Ala-pNA in buffer (20 mM Tris, 100 mM NaCl, pH 7.5); plates were briefly centrifuged to ensure that the solution is in the bottom of the wells. The plate loaded with the substrate solutions was pre-equilibrated for 20 min at 50 °C. Then, 10 $\mu$L of the protein solution (in the same buffer as the substrate) was added on each well in order to reach a final protein solution concentration on each well of 5 ng/$\mu$L. All solutions contained 2.8% (vol/vol) dimethylsulfoxide (DMSO; Sigma-Aldrich), which increases the solubility of the substrates. In order to minimize changes in the substrate solution (e.g., temperature) upon the protein addition, the plate was kept above the plate-reader thermostat and an electronic multichannel pipette was employed to load the protein solution into the wells and gently mix the solution. We estimated the pNA concentration from the solution absorptivity (molar absorption coefficient for the pNA at 410 nm of 8800 M$^{-1}$ cm$^{-1}$). The path length (0.375 cm) was estimated considering the shape and dimensions of the plate wells and the final volume of the solution. Before analysis, curves from blank sample (no protein) were substracted. The time-dependent absorbance values were analyzed with in-house written python scripts, by fitting the initial rate with a linear equation. Duplicate measurements (time traces of pNA absorbance) were performed. The error estimate of these initial slopes was obtained from the python function lmfit (least-squares fit routine). The difference of the duplicate measurements was small (ca. 3% or less), of the same order as the error estimate. These initial-regime slopes as a function of the substrate concentration were fitted to obtain Michaelis-Menten parameters K$_M$ and k$_{cat}$, reported in Figs. 1d and 4b,c and Supplementary Table 1. In this fit, all data points (including duplicates) were used in a joint fit. To determine the error estimates of the K$_M$ and k$_{cat}$ parameters, a Monte Carlo approach was chosen, following the principles described e.g., in ref. [80]. In brief, 1000 noisy data sets (initial slope vs. substrate concentration) were created, assuming a normal distribution around the experimentally obtained slopes with $\sigma$ corresponding to the error estimate of the slope (see above). The reported error bars in Fig. 4b and Supplementary Table 1 are the standard deviations over these 1000 Monte Carlo fits.

The measurements shown in Supplementary Figures 17 and 18 were done with essentially the same approach and minute changes: measurements on the same instrument as above were done in 96-well plates, equally at 50 °C. In all cases, the wells were filled with 80 $\mu$L of H-Leu-pNA substrate solution at 6.4 mM in either 20 mM Tris, 100 mM NaCl, pH 7.5, or in 100 mM CAPSO buffer at pH 9.3 or in 100 mM MES buffer at pH 5.3. To investigate the role of free histidine, 200 mM histidine solution was prepared in pH 7.5 buffer and the volume added to the reaction well was adjusted to have final concentration of 1, 5 or 20 mM histidine in the reaction solution.

**Bioinformatic analyses**. An initial seed for the co-evolution analysis was built using the sequences contained in the PFAM seed of the M42 Peptidase family (PFAM ID: PF05343) and aligned using the MAFFT utility. The alignment was then curated, removing overly gapped regions. This resulted in a sequence model consisting of 353 positions, covering the whole width of the *Pyrococcus horikoshii* TET2 peptidase (Uniprot ID O59196). A hmmer model of the family was then built using the hmmbuild utility and used to search the uniport database (union of TREMBL and Swissprot datasets, release 07_2019) for homologs using the hmmsearch utility, with standard inclusion thresholds. To remove fragments, the retrieved homologs were further filtered by coverage, keeping only sequences containing no more than 25% gapped positions. The loop region of TET was defined as lying between V120 and Q138 in the *Pyrococcus horikoshii* TET2 peptidase. Starting from this final Multiple Sequence Alignment (MSA), logo sequences considering only the mentioned loop region, including some neighboring residues due to highly conserved physicochemical properties between them (residues 115 to 139) were made using seqlogo[90], a method that takes the position weight matrix of a DNA sequence motif and plots the corresponding sequence logo according to parameters. Column heights in Fig. 4a are proportional to the information content. Regarding sequence identity, no significant differences were observed between logo sequences considering full MSA versus 90% sequence identity. The frequencies at the position corresponding to His123 in TET2 across the alignment are: His 88%, Gaps 7.2%, and the remaining AA all have frequencies of ≤1%. The conservation of the His is very strong for close homologs, and there are near regions with high levels of conservation too in the MSA, suggesting that the remaining sequences having gaps or other amino acids in this particular position correspond to remote homologs.

Direct-Coupling Analysis (DCA) was performed using the asymmetric version of the pseudo-likelihood maximization method, implemented in the lbsDCA code[88], using standard regularization parameters. To remove sampling bias, sequences were reweighted by identity, downweighing sequences with more than 90% sequence identity to homologs. DCA results were processed using utilities in the dcaTools package[88] (https://gitlab.com/ducciomalinverni/lbsDCA). To ignore uninformative very-short range predictions, all reported predictions and accuracies are for residue pairs separated by more than four residues along the chain. Structural contacts were defined by inter-atomic distances between heavy-atoms below 8 Å.

The sequence mining procedure resulted in the extraction of 26'067 TET homologs with at least 75% coverage. After reweighting by sequence identity, the number of effective sequences was of 9157.67, giving an excellent B$_{Eff}$/N$^{Pos}$ ratio of 25.9, where B$_{Eff}$ denotes the number of effective sequences after weighting sequences by sequence identity[91]) and N$^{Pos}$ denotes the number of residue positions (i.e., columns) in the MSA. DCA prediction benchmarked on the 1Y0Y structure show excellent prediction accuracies over a large range of predictions (Supplementary Figure 10a). Notably, considering the top 2 N = 706 highest ranked DCA predictions results in a prediction accuracy of 88%. Ignoring the false-positives rising from predictions falling in regions where the PDB structure is not defined, the accuracy rises above 90%. Inspection of the predicted contacts with respect to the 1Y0Y PDB structures (Supplementary Figure 10b and Fig. 3) highlights the prediction of multiple sets of contacts involving the loop region. These can be separated in a set formed by loop-loop interactions, a set of putative intra-molecular loop contacts, and a third set of putative inter-molecular loop interactions (Supplementary Figure 10). Supplementary Table 2 reports the list of all 19 predicted contacts involving the TET loop.

**Reporting summary**. Further information on research design is available in the Nature Research Reporting Summary linked to this article.

## Data availability

Source data are provided with this paper: REDOR data and the analysis routines (GAMMA simulation program and python analysis script), $^{13}$C relaxation data, the Direct-Coupling Analysis (co-evolution) data, the activity assay data, the $^{15}$N relaxation data (Fig. S3) and the MD-derived contact data (Fig. S11). Data have also been deposited on Mendeley Data, https://doi.org/10.17632/vx2xmjgmk9.3. The Protein Data Bank (PDB) entries of the structures used in this work are: 2WZN, 2CF4, 1Y0R, 2WYR, 6R8N, 6F3K, 1Y0Y. The plasmid for expressing TET2 is deposited at Addgene under accession number 182428. Other data are available from the authors upon request. Source data are provided with this paper.

## Code availability

Python and GAMMA code for the fit of the REDOR data is available on Mendeley Data, https://doi.org/10.17632/vx2xmjgmk9.3 and has been provided with the manuscript.

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

## Acknowledgements

We are grateful to Bernhard Brutscher, Alicia Vallet, and Adrien Favier for excellent NMR platform operation and management. The plasmid coding for TET2 was kindly provided by Bruno Franzetti and Jerome Boisbouvier (IBS Grenoble). We thank Anne-Marie Villard and the RoBioMol platform for preparing the loop deletion construct. The RoBioMol platform is part of the Grenoble Instruct-ERIC center (ISBG; UAR 3518 CNRS-CEA-UGA-EMBL) within the Grenoble Partnership for Structural Biology (PSB), supported by FRISBI (ANR-10-INBS-0005-02) and GRAL (ANR-10-LABX-49-01), financed within the University Grenoble Alpes graduate school (Ecoles Universitaires de Recherche) CBH-EUR-GS (ANR-17-EURE-0003). This work was supported by the European Research Council (StG-2012-311318-ProtDyn2Function to P. S.) and the French Agence Nationale de la Recherche (ANR), under grant ANR-14-ACHN-0016 (M.P. and A.B.).

## Author contributions

D.F.G. performed and analyzed enzymatic assays, collected and analyzed NMR data, prepared figures and contributed to writing the manuscript. P.M. and H.F. prepared protein samples. D.M. performed the co-evolution analysis. M.P. and A.B. performed and analyzed MD simulations. I.S. performed activity assays. A.H. produced protein samples. J.P.B. prepared the logo plots and underlying sequence alignment. P.S. designed, initiated, and supervised the study, performed and analyzed NMR experiments, prepared the figures, and wrote the manuscript with input from all co-authors.

## Competing interests

The authors declare no competing interests.
