## [Peer Review File · Nature Communications]

Reviewer comments, first round

Reviewer #1 (Remarks to the Author):

Paul Schanda and colleagues present an integrative study of the flexible loop in the TET2 aminopeptidase. They used mostly solid-state NMR in their research but combined it in a sophisticated manner with results from MD simulations and co-evolution analysis. Lastly, they tested - and proved - their hypotheses with functional assays. TET2 aminopeptidase is a large and challenging system. The authors convincingly demonstrate that the flexibility of the loop is crucial for efficient enzymatic activity. I really appreciate this quite comprehensive investigation and believe that it would be an excellent fit to the scope and broad readership of Nature Communications. The data presented are of highest quality, the paper is overall well written, and the authors made a great effort to combine different experimental and theoretical/simulation data in order to shed light on the structure, dynamics, and function of the previously enigmatic loop. While I strongly support the acceptance of this outstanding work, I have a few issues that should be addressed before acceptance:

- In the introduction it is stated that "TET2 is ... the largest structure solved to date" I find this statement too general. It is the largest structure solved in which sense? By solid-state NMR? By an integrated approach? What about the recent cryo-EM/NMR study of Zinke et al. (Nat. Comm. 2020) e.g., where the structure of >40 rings times 6 subunits were solved - without a previous crystal structure?
- The INEPT-based hNH spectrum is not shown ("data not shown"). This should be done in the SI. There are more cases of this and in general this should not be allowed.
- In Fig.3 some panels appear to be exchanged (please check c and d and what is written in the figure legend).
- "Collectively, three ... approaches, reveal large amplitude motion ..., which we show to occur on a time scale of 10 us to 1 ms". This comes from NMR, but not from MD and co-evolution analysis, does it? Please describe better, where the time scale comes from. The whole paragraph ("Co-evolution and MD simulations...") is a bit short and could provide more details.
- A similar problem as in Fig. 3 also occurs in Fig. 5 (b and c).
- The material that was uploaded to the IST repository was not available to the reviewer. It should be rather provided as SI files.
- The first part of the discussion is fine but the second part claims a generality of the approach that is maybe not justified. A very high resolution in the solid-state NMR spectra is required and that may not always be the case.
- One thing to note is also that most residues of the loop are actually not observed in the NMR spectra and that while the obtained relaxation dispersion data are certainly highly interesting this approach should not be oversold in the last paragraph.

Reviewer #2 (Remarks to the Author):

The authors investigate the mobility of a loop in Tet2 following their earlier paper from 2019 in Nature Communications and much earlier work from the Groll group. In their earlier paper, a structure was published based on EM and NMR data. Now the mobility of an ill-defined loop with conserved residues is discussed and an implication of its residues in substrate 'fixation'. As measures of mobility methyl group relaxation times and associated order parameter are presented. I did not see any measurements of amide signal relaxation. The authors focus in particular on the behaviour of a PPH motif at the beginning of the flexible loop. The paper contains

some useful kinetic data about variants and of course the methyl relaxation is interesting, but its interpretation is 'surprising'.

I cannot recommend publication in Nature Comm since the presentation is falling short in discussing the matter adequately. First of all, it is claimed that the ill-defined loop starting with VPPH is very mobile, and the authors suggest by means of Figure 1 that it reaches from V120 to Q138. As indicators of mobility the S^2 of methyl groups of V120 and I139 are presented. However, V120 is still close to reasonably defined in the Groll structures, and I139 is by no means disordered according to the existing structures which include a helix involving the stretch 135DWDQI139 in Tet2 or of related residues in other proteins. So, the measurement of methyl order parameters does not report on loop mobility, but on adjacent residues. What is not said in the introduction or discussion sections is that the motif PPH is treated as defined in the structures of the sufficiently homologous Tet1, Tet3 and Tet4 (e.g. 2wyr, 2wzn, 2cf4) and the (absolutely?) conserved H is part of the active center of another protomer in the oligomer and thus most likely not related to substrate recognition/fixation. In those structures there are fewer loop residues ill-defined or even none. I agree, the b -values of the PPH motif are high and close to too high, but a study that would match the presented claims would look at the dynamics of the histidine directly. Worth of note, the W following the PPH motif seems to be reasonably ordered, the helix mentioned above is ok. The authors did not check on proline trans/cis isomerisation.

The introduction suffers from not putting emphasis on necessary information as mentioned above. It lacks details to TET proteases that are required to have at hand for understanding the presented data. Also, it is often too general. The paper needs total rewriting, with a proper discussion of the state-of-the-art in Tet protease structural biology. Whereas I would cut on general statements and shorten the apologetic sentences on why solution NMR is not applied (there are more techniques that are not applied), I miss a careful introduction into the structure of the active center of TET proteases, the particular enzyme kinetics, a discussion of the different proposed mechanisms, potential similarities of the active center to those of other proteases, a statement as to where the binding site of the antagonist (amastatin) is situated, how far from the active center, its mode of action, the binding constant, all other relevant kinetic constants, a word more to specificity, etc.

Reviewer #3 (Remarks to the Author):

The manuscript by Gauto et al studies the effect of a flexible loop on the activity of the large peptidase TET2. They use a beautiful combination of structural and bioinformatics techniques to unravel how this loop contributes to TET activity. The work has been done with great care and is an excellent demonstration how a combination of approaches can resolve challenging questions in structural biology. The paper is particularly well and clearly written (Congratulations!) and Figures are generally very nice (one exception see below). The Supplementary Material is of high quality. The manuscript fits particularly well into Nature Communications in terms of impact, scope and quality.

I see however one main weakness residing, namely that the analysis of the functional role of residue Histidine 123 is just not fully completed:

The authors show that the conserved Histidine residue 123 plays a crucial role in the catalytic mechanism of TET and that it is the function of the loop to bring the Histidine somehow into interaction with the substrate. Regarding the functional role there, the authors propose some generally stabilizing effects. But this may not be the complete answer. I could very well imagine that the Histidine 123 has a more specific effect that is overlooked in the current form of the manuscript.

One reason for this suspicion is that the residue is fixed by evolution in a specific position. If the contacts to substrate were rather unspecific, it should be possible to have the Histidine also on one of the nearby sites and it would then be not as strictly conserved as it is. Secondly, the adjacent Pro-Pro motif stiffens the loop and thus restrains the His to certain motions, which might indicate a more specific role that benefits from a certain steric orientation. Thirdly, His 123 is not positioned at the tip of the loop, where it would have the largest reach to interact with the substrate, but it is located at the flank of the loop, where it has a reduced action radius.

The Histidine might thus have a more specific role in the mechanism of TET than currently suggested. For example, it might interact with the transition state of the cleavage reaction, or the Zn ions directly or indirectly, but in a specific manner. I propose the following analyses and experiments to get better insights:

(1) Structural modelling, where exactly the His can sterically reach on TET and on a substrate, also as a function of the presence of either Pro-Pro / Ala-Ala / Gly-Gly motif adjacent to it. The results from this exploration might contain strong hints on the specific role of the Histidine.

(2) Theoretical considerations from the perspective of the chemistry of the peptidase reaction, whether a Histidine might play a role in stabilizing the known or expected transition state of the reaction, and then, with the result from (1), whether His123 can reach that place.

(3) Experimental exploration, to which alternative positions the Histidine can be moved in the loop. Is it strictly residing at position 123 or can it be moved to other places, e.g. 124, 125, etc? What is the effect of such move on activity?

(4) Measurement of protein activity in the H123Y mutant, but in the presence of an excess of free Histidine in the buffer. This might also give some hints on the functional role of the Histidine 123.

Minor point:

- All Figures are very nice except Figure 3, which is somewhat hard to access.

(1) Generally, the Figure is very crowded and packed and many details are too small. This is a very important Figure and it would be great, if the authors could make an effort to improve it.

(2) There is some confusion between the Figure legends and the content of panels c and d.

Unclear, which is the MD and which the co-evolution.

(3) The position and orientation of the subunit shown in panel b inside the whole complex remains unclear to me, even though some zoom-in is shown.

(4) The Figure legend of panel c should state what is defined as a contact in the MD.

Reviewer #1 (Remarks to the Author):

Paul Schanda and colleagues present an integrative study of the flexible loop in the TET2 aminopeptidase. They used mostly solid-state NMR in their research but combined it in a sophisticated manner with results from MD simulations and co-evolution analysis. Lastly, they tested - and proved - their hypotheses with functional assays. TET2 aminopeptidase is a large and challenging system. The authors convincingly demonstrate that the flexibility of the loop is crucial for efficient enzymatic activity. I really appreciate this quite comprehensive investigation and believe that it would be an excellent fit to the scope and broad readership of Nature Communications. The data presented are of highest quality, the paper is overall well written, and the authors made a great effort to combine different experimental and theoretical/simulation data in order to shed light on the structure, dynamics, and function of the previously enigmatic loop. While I strongly support the acceptance of this outstanding work, I have a few issues that should be addressed before acceptance:

We thank the reviewer for this very positive assessment, and for the time they have taken for this careful study and the questions below.

- In the introduction it is stated that "TET2 is ... the largest structure solved to date" I find this statement too general. It is the largest structure solved in which sense? By solid-state NMR? By an integrated approach? What about the recent cryo-EM/NMR study of Zinke et al. (Nat. Comm. 2020) e.g., where the structure of >40 rings times 6 subunits were solved - without a previous crystal structure?

We have removed this statement. Our reasoning was the following: while there are indeed solid-state NMR structures of assemblies of essentially infinite size (amyloids, or the tubes by Zinke *et al.*), the subunit size in all these systems was much smaller. The challenge when it comes to protein structure determination by NMR is primarily the monomer size. In this sense, TET2 with 12 x 39 kDa stands out. But we prefer to remove the statement because it does not add that much to the present paper.

- The INEPT-based hNH spectrum is not shown ("data not shown"). This should be done in the SI. There are more cases of this and in general this should not be allowed.

We indeed had 4 instances where we omitted showing background data in the original submission. We have now addressed these instances as follows.

1. In the results section we had mentioned the comparison of INEPT-based hNH spectrum and CP-based hNH spectrum, but had not shown it. Now we added this figure (Fig. S2).
2. In the methods section, we had: ^{13}C - ^{13}C spectra of MPD-precipitated, isopropanol-precipitated and sedimented samples were highly similar," and we now added a new Figure that shows these spectra (Fig. S21).
3. We had written that "V120 is ca. 6 - 18 Å away from the nearest atom of the inhibitor in the MD ensemble (Fig. 5c and data not shown)". We think that Figure 5c is actually explicit enough to show what we want to show, and we decide, in the interest of readability of the Supplementary Material, not to add any additional figure. The point we want to make is that we can safely neglect direct chemical-shift changes of V120 upon inhibitor-binding at the active site. Figure 5c shall make this sufficiently clear, we think. Thus, we simply dropped the statement "and data not shown".
4. In the last instance where we had "data not shown" in the original manuscript, we referred to a multiple sequence alignment: "Regarding sequence identity, no significant differences were observed between logo sequences considering full MSA versus 90% sequence identity (data not shown)." Again for reasons of readability of the Supporting Information, we simply drop the "data not shown" statement. The reason for this is that it would not be straightforward to show this in the Supp. Info., and on the other hand the sequences are all in public data bases, and can be retrieved by anyone.

- In Fig.3 some panels appear to be exchanged (please check c and d and what is written in the figure legend).

This is right, and we thank Reviewer 1 and Reviewer 3 to highlight that we got the order of the figure caption of (c) and (d) wrong. We have modified Figure 3, according to Reviewer 3's comments, and corrected this figure panel order.

- "Collectively, three ... approaches, reveal large amplitude motion ..., which we show to occur on a time scale of 10 μs to 1 ms". This comes from NMR, but not from MD and co-evolution analysis, does it? Please describe better, where the time scale comes from. The whole paragraph ("Co-evolution and MD simulations...") is a bit short and could provide more details.

The reviewer is right that the time scale comes from MAS NMR only. We have re-written this statement as follows: "Collectively, three fundamentally different approaches, MAS NMR, MD and co-evolution analysis, reveal large amplitude motion of the functionally important loop. *The MAS NMR relaxation-dispersion data show that this process occurs on a time scale of ca. 10 μs to 1 ms.*"

- A similar problem as in Fig. 3 also occurs in Fig. 5 (b and c).

Thank you, we have corrected the order of the caption labels.

- The material that was uploaded to the IST repository was not available to the reviewer. It should be rather provided as SI files.

Thanks for pointing out this issue. We had a problem with the link to the deposition of the data. Now, we have uploaded a pretty substantial data set to an open access repository.

The deposited data comprise:

methyl dynamics data (R1, R1rho/NERRD, REDOR
direct coupling analysis and MSA

The data can be retrieved under these DOI number: <https://doi.org/10.17632/vx2xmjgmk9.1>

(the multiple sequence alignment is in version 2: <https://doi.org/10.17632/vx2xmjgmk9.2>)

The files contain many text files, python programs, C++ programs, figures, and are difficult to provide as SI files, to our understanding.

- The first part of the discussion is fine but the second part claims a generality of the approach that is maybe not justified. A very high resolution in the solid-state NMR spectra is required and that may not always be the case.

We reply to this point and the next point below:

- One thing to note is also that most residues of the loop are actually not observed in the NMR spectra and that while the obtained relaxation dispersion data are certainly highly interesting this approach should not be oversold in the last paragraph.

These comments on the generality (or lack of generality) are valid and while we think that the methods we used here are very useful in a range of applications, we do not want to hide the requirements and potential weaknesses. We propose to make the following changes in the Discussion section:

We removed: "Our study demonstrates that such quantitative dynamics experiments by MAS NMR, combined with other biophysical methods, can be decisive to link static structures to functional mechanisms."

We replaced it by: "The prerequisite for performing such studies at the atomic level is that the individual cross-peaks are visible and resolved. In the present case, the spectral resolution is, generally, very high, even at modest magnetic field strength (600 MHz ¹H Larmor frequency). While similarly high resolution has been reported for a number of other systems by MAS NMR, the present approach is not necessarily general for any system. Also, due to the extensive μ s mobility, most of the backbone of the key loop here went undetected. In the general case, a combination of methods - MAS NMR, solution-NMR and other spectroscopic and structural methods along with simulations may be required, depending on the molecular system. Such insight into dynamics can be decisive to link static structures to functional mechanisms."

Reviewer #2 (Remarks to the Author):

The authors investigate the mobility of a loop in Tet2 following their earlier paper from 2019 in Nature Communications and much earlier work from the Groll group. In their earlier paper, a structure was published based on EM and NMR data. Now the mobility of an ill-defined loop with conserved residues is discussed and an implication of its residues in substrate 'fixation'. As measures of mobility methyl group relaxation times and associated order parameter are presented.

We thank the reviewer for the careful reading and critical evaluation of the manuscript. New experiments and re-writing/extension of the manuscript, based on their comments, was done as outlined below, and have, we think, improved the manuscript.

We had been aware of the work from the research groups of Prof. Groll, Dr. Franzetti and Prof. Dutoit (on TETs) as well as others (on homologs), and have now provided more reference to this work, in the Introduction section.

I did not see any measurements of amide signal relaxation.

The data that show amide signal relaxation are presented in Figure S3 (which was Figure S2 in the original submission). It indeed shows that I135 undergoes microsecond dynamics, more than any other residue in the protein that we detected. This finding shows that the backbone of this region undergoes microsecond dynamics. We are aware that this residue is not central to this loop and is defined in the crystal structure, but it is adjacent and points to even more dynamics of the loop. We discuss this below.

The authors focus in particular on the behaviour of a PPH motif at the beginning of the flexible loop. The paper contains some useful kinetic data about variants and of course the methyl relaxation is interesting, but its interpretation is „surprising“.

I cannot recommend publication in Nature Comm since the presentation is falling short in discussing the matter adequately. First of all, it is claimed that the ill-defined loop starting with VPPH is very mobile, and the authors suggest by means of Figure 1 that it reaches from V120 to Q138. As indicators of mobility the S squared of methyl groups of V120 and I139 are presented. However, V120 is still close to reasonably defined in the Groll structures, and I139 is by no means disordered according to the existing structures which include a helix involving the stretch 135DWDQI139 in Tet2 or of related residues in other proteins. So, the measurement of methyl order parameters does not report on loop mobility, but on adjacent residues.

We perfectly agree with the reviewer on their assessment of the location of these residues.

We have made clearer at the beginning of the results section, which are the residues that are not modeled in the crystal structure:

“The catalytic chamber of TET2 comprises twelve long loops, one from each subunit. [removed:., comprising residues 120-138.] Interestingly, in 3D structures of TET2 obtained by crystallography ¹(Borissenko2005), these loop regions have not been modeled [added: (residues 120-132 are missing)].”

We had written in our original submission that the two residues (120, 139) are at the very beginning and end of the loop, and that the loop dynamics may well be significantly larger, in its center, than what these two reporter residues “see”. Here is what we had in the original submission:

“The well-resolved cross-peaks of Val 120, located toward the beginning of the loop, and of Ile 139, located at the C-terminal junction of the loop to a β -strand...” (in the section “A highly dynamic loop in the catalytic chamber controls enzyme activity” or the Results section).

To make it clearer that, as the reviewer points out, the two residues are not centrally located in the loop, that Ile 139 is part of a single-turn helix, and that the loop motions may be actually underestimated by the two probes, we have added the following sentence:

“It is noteworthy that the two probes may underestimate the full loop mobility: Val 120 is not too far from a residue that still had been modeled in the crystal structure, although with high B-factor (Ser 119); Ile 139 is located in a part that had been modeled in the crystal structure. Thus, while the two probes clearly have the largest amplitude among all methyl-bearing residues we observed, with order parameters of the order of 0.2, the actual loop motion may be even larger.”

Of note, we actually have a third probe that we quantitatively analyzed, D135, as we point out in the second answer to this reviewer (above). Moreover, the fact that we were unable to assign the majority of this loop, whereas the large majority of residues could be identified in our multi-dimensional NMR spectra, is a further sign for motion of the loop.

We now have new data further supporting our assessment that the loop is highly dynamic. We have collected a 1H-13C spectrum of a sample in which position 123 carries a specifically labeled (and generally highly sensitive) 1H-13C pair, and fail to detect the corresponding correlation peak. This is discussed below (-> new Figure S4).

We want to stress here that the actual amplitude of the motion is not overly important, in the sense that if we had a residue right in the center of the loop, and it had an order parameter of 0.1 or 0.05 (which is possible), it would not change any of our conclusions. The fact that the probes that we report (V120, D135, I139) already show large-amplitude microsecond dynamics, together with the MD results, and the fact that the loop is in very different positions in different structures (->Fig. S1, S13) and that at least part of it are not modeled in most of the TET structures establish that this loop is highly flexible in TET proteases.

What is not said in the introduction or discussion sections is that the motif PPH is treated as defined in the structures of the sufficiently homologous Tet1, Tet3 and Tet4 (e.g. 2wyr, 2wzn, 2cf4) and the (absolutely?) conserved H is part of the active center of another protomer in the oligomer and thus most likely not related to substrate recognition/fixation.

We politely disagree with the reviewer's assessment that "the conserved histidine is part of the active center of another protomer". In fact, the His is quite far from the active site in those crystal structures where it has been modeled.

To make this point clear, we have prepared a figure that shows the position of this histidine in the three structures that the reviewer mentions, as well as others. The distance between the His and the zinc site can indeed be as close as ca. 4 Å in one case (2wyr, TET1), but is ca. 9 Å (2cf4, TET1), >12 Å (2wzn, TET3) >13 Å (1vhe, a TET homolog from *B. subtilis*), and up to over 20 Å (6r8n, TET2 – this is the cryo-EM structure for which electron density is clearly visible for the loop, as shown in Figure S1i). We furthermore measured the distance in a homologous M42 protein that assembles as a dodecamer, from *Desulfurococcus kamchatkensis* (PDB: 4wwv) and find a distance > 12 Å; in PfTET4 (4x8i) the distance is also beyond 11 Å.

In another homolog with the same fold, PDB 3kl9 from *Streptococcus pneumoniae*, the loop is not modeled in the structure, just like in TET2, i.e. residues 117-131 are absent.

We do not see how these structures would provide any hint that this histidine is part of the active center, as the reviewer proposes.

We thank the reviewer for making us look into this question additionally, and write more arguments into the paper to support our view that the loop is highly flexible and the His is not part of the active site in the crystal structures.

In those structures there are fewer loop residues ill-defined or even none.

We agree that the number of residues that have not been modeled vary in these structures.

To be precise: the numbers of residues not modeled in the above structures in this loop region:

2wyr: 5 residues not modeled

2cf4: 0 residues not modeled (but high B-factors, see Figure S1h)

2wzn: 9 residues not modeled (high B-factors in the surrounding residues, Figure S1h)

1vhe: 0 residues not modeled (but B-factors are ca 45 Å² for this region, while it is ca. 5-10 for most of the protein)

4 wwv: 4 residues not modeled

3kl9: 14 residues not modeled

1y0r (that's TET2): 12 residues not modeled

We do think that we have a clear case here that this loop is highly dynamic.

We have looked it also from a different angle, namely MD simulations, see a few lines below.

We want to put forward here another argument why we are convinced that this loop is dynamic, and why the dynamics is most likely a common theme in this family: the co-evolution data. These data show that there is co-evolution between residues in the loop and residues elsewhere in the chamber. Importantly, a rigid structure (such as the few X-ray structures where the loop is modeled) cannot fulfill these evolutionarily conserved contacts. In other words: the loop must be dynamic to fulfill the contacts.

I agree, the b-values of the PPH motif are high and close to too high, but a study that would match the presented claims would look at the dynamics of the histidine directly.

As described in the manuscript, using several NMR experiments, the extensive microsecond dynamics makes large part of the residues in this loop unobservable.

Previously, we had not directly shown whether residue 123 is observable or rather broadened due to the microsecond dynamics.

In the revised manuscript, we provide new experimental data to look into this question: we have recently developed and demonstrated a method that allows the isotope labeling of phenylalanine residues at high sensitivity and resolution (Gauto et al & Schanda, JACS 2019, *J. Am. Chem. Soc.*, 141(28), 11183–11195). We reasoned that with this very sensitive method, and given the sparsity of a spectrum that contains only the para-CH sites of Phe residues, we shall be able to see a Phe residue in the loop. The only reason why we would not see it, we reasoned, is if it undergoes large-scale microsecond motion that would broaden the resonance. We, thus, used the H123F mutant and labeled it accordingly. Maybe disappointingly, but definitely in agreement with the rest of our data, we were not able to see an additional peak. Thus, there is nothing we can do to directly detect this position with the tools we have available.

This new figure with the Phe spectrum is now shown in Figure S4.

We have, however, addressed the implicit assumption by the reviewer that the loop is actually not disordered in the other TET structures, using microsecond-long MD simulations of the dimers of TET1 and TET3, in addition to TET2, and we have done a simulation also of the dimer of TET2 – in addition to the dodecamer of TET2, which we had simulated before.

(The computational effort to perform these simulations is part of why our revision has taken a long time.)

We have prepared a new figure showing these simulation results (Figure S12).

Worth of note, the W following the PPH motif seems to be reasonably ordered, the helix mentioned above is ok. The authors did not check on proline trans/cis isomerisation.

We assume that the reviewer refers to a direct experimental analysis of the cis/trans polymerization reaction (kinetics, thermodynamics). As outlined before, the microsecond dynamics makes direct observation of these residues impossible. Moreover, prolines are notoriously difficult to detect because of the lack of an amide proton. However, we have investigated the conformation of the prolines, and in all structures we investigated we find the prolines in trans conformation. We have noted this for a few cases, including the ones that the reviewer mentioned, and made a note in Figure S2.

The introduction suffers from not putting emphasis on necessary information as mentioned above. It lacks details to TET proteases that are required to have at hand for understanding the presented data. Also, it is often too general. The paper needs total rewriting, with a proper discussion of the state-of-the-art in Tet protease structural biology. Whereas I would cut on general statements and shorten the apologetic sentences on why solution NMR is not applied (there are more techniques that are not applied), I miss a careful introduction into the structure of the active center of TET proteases, the particular enzyme kinetics, a discussion of the different proposed mechanisms, potential similarities of the active center to those of other proteases, a statement as to where the binding site of the antagonist (amastatin) is situated, how far from the active center, its mode of action, the binding constant, all other relevant kinetic constants, a word more to specificity, etc.

We have substantially re-written the introduction, performed new experiments and added those to the Results section, and we changed the Discussion section. As one can see from the highlighted changes in the paper, we have substantially re-worked the text.

In particular, we include now in the introduction section the following aspects, as proposed by the reviewer:

- We have described in some detail the active site, the chelating residues and the activation of a water molecule in the catalytic mechanism
- We have described how the inhibitor binds, and shown the details of the active site and inhibitor binding in Figures S1 and S20.
- We have mentioned and cited kinetic rate constants (which are in agreement with ours).

In addition, the results and Discussion section is extended to discuss the possibility of a direct involvement of the His, which our new enzyme-activity measurements have opened.

Reviewer #3 (Remarks to the Author):

The manuscript by Gauto et al studies the effect of a flexible loop on the activity of the large peptidase TET2. They use a beautiful combination of structural and bioinformatics techniques to unravel how this loop contributes to TET activity. The work has been done with great care and is an excellent demonstration how a combination of approaches can resolve challenging questions in structural biology. The paper is particularly well and clearly written (Congratulations!) and Figures are generally very nice (one exception see below). The Supplementary Material is of high quality. The manuscript fits particularly well into Nature Communications in terms of impact, scope and quality.

We thank the reviewer for the extraordinarily positive feedback above, as well as the thoughtful suggestions and ideas below, that we have addressed as described in the following.

I see however one main weakness residing, namely that the analysis of the functional role of residue Histidine 123 is just not fully completed:

The authors show that the conserved Histidine residue 123 plays a crucial role in the catalytic mechanism of TET and that it is the function of the loop to bring the Histidine somehow into interaction with the substrate. Regarding the functional role there, the authors propose some generally stabilizing effects. But this may not be the complete answer. I could very well imagine that the Histidine 123 has a more specific effect that is overlooked in the current form of the manuscript.

One reason for this suspicion is that the residue is fixed by evolution in a specific position. If the contacts to substrate were rather unspecific, it should be possible to have the Histidine also on one of the nearby sites and it would then be not as strictly conserved as it is. Secondly, the adjacent Pro-Pro motif stiffens the loop and thus restrains the His to certain motions, which might indicate a more specific role that benefits from a certain steric orientation. Thirdly, His 123 is not positioned at the tip of the loop, where it would have the largest reach to interact with the substrate, but it is located at the flank of the loop, where it has a reduced action radius.

The Histidine might thus have a more specific role in the mechanism of TET than currently suggested. For example, it might interact with the transition state of the cleavage reaction, or the Zn ions directly or indirectly, but in a specific manner. I propose the following analyses and experiments to get better insights:

(1) Structural modelling, where exactly the His can sterically reach on TET and on a substrate, also as a function of the presence of either Pro-Pro / Ala-Ala / Gly-Gly motif adjacent to it. The results from this exploration might contain strong hints on the specific role of the Histidine.

(2) Theoretical considerations from the perspective of the chemistry of the peptidase reaction, whether a Histidine might play a role in stabilizing the known or expected transition state of the reaction, and then, with the result from (1), whether His123 can reach that place.

(3) Experimental exploration, to which alternative positions the Histidine can be moved in the loop. Is it strictly residing at position 123 or can it be moved to other places, e.g. 124, 125, etc? What is the effect of such move on activity?

(4) Measurement of protein activity in the H123Y mutant, but in the presence of an excess of free Histidine in the buffer. This might also give some hints on the functional role of the Histidine 123.

These are very good suggestions which triggered new experiments and analyses on our side.

We have performed additional enzymatic assays in the presence of free histidine in solution (point (4)). These experiments show that in the presence of histidine the activity actually drops sharply (in contrast to the reviewer's suggestion). We speculate that it binds to the active site. (-> Fig. S 17).

However, we have done other experiments, not suggested by the reviewer, which we find very interesting: we measured the activity at different pH values, and see that in the His-free mutants, but not in the WT protein, the activity increases at higher pH. The former may appear normal: the reaction is catalyzed by OH⁻ ions. But why is WT protein already as active at pH 7.5 as at pH 9.3? We propose that this has to do with a role of the His in the proton-abstraction from water. We propose that it assists the proton abstraction (which is "stored" on E212). Thus, the loop may have a dual role: substrate stabilization and a more "chemical" activity.

We have also done structural modeling (point (1) above). Figure S20 answers the question whether His123 may reach the active site: yes, it can. It can actually both interact with the substrate and possibly reach into the active site (thus contributing to proton-abstraction from water).

A new figure, S19, also shows that in other peptidases there are histidines that interact with bound ligand/substrate, and we propose that such interactions may be the ones that explain the role of H123 in TET2.

We thank the reviewer for proposing this additional direction. We have added these conclusions to the end of the Results section, and the Discussion section.

Minor

point:

- All Figures are very nice except Figure 3, which is somewhat hard to access.

(1) Generally, the Figure is very crowded and packed and many details are too small. This is a very important Figure and it would be great, if the authors could make an effort to improve it.

We have redone this figure largely (point (3) below).

(2) There is some confusion between the Figure legends and the content of panels c and d. Unclear, which is the MD and which the co-evolution.

Yes, this is right, and the panel annotation was erroneous. We have resolved this confusion now.

(3) The position and orientation of the subunit shown in panel b inside the whole complex remains unclear to me, even though some zoom-in is shown.

We have now made more effort to show the exact orientation of the subunits, before zooming into it. Please see the new panel 3c. We also made changes to panel (b), namely we have made the outside-view larger. Furthermore, we made legends into the figure, to make clear what the red spheres mean.

I hope that this figure is now more readable.

(4) The Figure legend of panel c should state what is defined as a contact in the MD.

This is a good point. We had it in the Methods, but not in the figure caption. We now added to the caption: "Residues where considered in contact when the minimum inter-residue distance between any heavy atoms was below 5 Å."

Other changes we made to the manuscript:

We have changed the title to “Functional control of a 0.5 MDa TET aminopeptidase by a flexible loop revealed by MAS NMR”. We think that having the primary technique (MAS NMR) in the title will make the paper more discoverable by the MAS NMR community; given that a number of fairly novel MAS NMR approaches are presented, we find this useful.

We have made several small changes in the text and corrected a few typos. These changes are highlighted in the text.

We have updated Figure S11A (contacts observed in the MD trajectory). The former version was using a definition of the loop residues that is somewhat different.

Reviewer comments, second round

Reviewer #1 (Remarks to the Author):

The manuscript is now ready for acceptance.

Reviewer #2 (Remarks to the Author):

The authors have done a very good job in revising the manuscript that is now acceptable and surely very instructive to read.

However, since the conserved sequence PPHI/L is very important, I would still recommend to be more precise in the first sentences of the results section. It should be indicated which residues (often the PPHI/L motif) around the flexible ones of the loop are modelled in the different structures (including also 2cf4, maybe more) and particularly in 2wyr and 2wzn, further where the residues are situated within the structure (and with respect to the closest zinc ions of symmetry-related protomers), and which B-factors they have. It is not utterly correct to state 'high B-factors' for cases of 'somewhat higher B-factors'. 45 is still a number that makes the coordinates reliable. The B-factors for those modelled residues should be mentioned instead of a global and not quite correct statement. In the statement about MD simulations and the mobility of the loop regions it should be said which residues in which structure are mobile or not, and especially what is found for the PPH motif. This does not need many sentences and should in the ideal case be perpetuated through the text and especially considered in the discussion.

Reviewer #3 (Remarks to the Author):

The authors have revised the manuscript substantially and made several new experiments, including some of my prior suggestions. As mentioned before, the quality of the experiments and documentation is excellent and from a technical perspective, the work is at the international top level. The manuscript is well written and very interesting.

I have to admit I got a bit irritated by the massive insertions of background knowledge the authors needed to make into their introduction as a response to the comments by referee 2. I got the impression that parts of the background knowledge on this class of metalloproteases may have not been at the awareness of the authors at the initial submission. Still, I find their response to referee 2 and the connected discussion of other work in the field reasonable.

This is the first application of solid state NMR to this class of proteins and therefore definitely brings a new perspective into the field. I continue to support publication of the work in Nat Commun.

LEGEND

Responses in bold **black**

Changes to the manuscript in **blue**

Location of inserted text in (parenthesis)

Reviewer #1:

The authors have generally done an excellent job of addressing the points raised by all three reviewers. They have gone to significant effort and their responses are thorough. I think the article benefits from the inclusion of the ebola & novovirus datasets, which broaden the potential utility of Variabel, and the analysis of synthetic controls. I am therefore happy to support publication of the paper once a final issue is addressed:

R1Q1: I previously suggested that the authors should implement their proposed large-scale database/framework from public SARS-CoV-2 data, as the availability of such a resource would greatly improve the utility of Variabel for SARS-CoV-2 genomics. In their response, the authors have explained very convincingly why this is not feasible and I tend to agree. However, I don't think it is fair to propose this solution in the manuscript without also identifying the numerous major hurdles that the authors have provided in their response, which will most likely ensure that this solution is never implemented.

This is the main section that I'm referring to:

"Based on a simple simulation (see Figure 4B) we calculate that approximately 10,000 samples would be required to recover most of the intra-host variants if we assume variants occur randomly along the genome of SARS-CoV-2. Similarly, we also expect a small drop in performance of Variabel if the time series data included fewer samples (e.g., 2-5). Both scenarios could be improved by leveraging a centralized data depository of low frequency SNV for SARS-Cov-2. Follow-up studies can then leverage this resource to assess and evaluate the biological importance of observed low frequency variants within and across hosts over time. However, established COVID databases such as GISAID are limited only to consensus level sequences 30,31, which might be a limiting factor going forward in this or future pandemic or outbreaks."

In my opinion, the authors should: (1) either remove the suggestion that a centralized repository could solve this issue for Variabel or very clearly explain why they have not implemented this solution themselves and; (2) make it very clear to the reader that, without this large and complex resource, the detection of low-AF variants in small-scale cross-patient SARS-CoV-2 datasets will not be very accurate.

We appreciate the reviewer's valuable feedback. We have added the following text in the manuscript to emphasize that the detection for low allele frequency variants for small

scale cross-patient dataset will be limited without a centralized repository for low frequency SNVs for SARS-Cov-2.

> (Main Text, Page 6)

Given established viral genomic databases such as GISAID are limited only to consensus level sequences, coordinated community efforts to store and track low frequency variation across vast collections of SARS-CoV-2 datasets will significantly boost Variabel's ability to detect low-AF variants in cross patient samples.

Reviewer #2:

The authors have made a number of very useful additions to the revised manuscript including expanding the breadth of their analysis to include ebola and norovirus datasets.

R2Q1: While the underlying results appear robust, the choice of reference sequence for the norovirus dataset is not ideal. The prototype NCBI strain is genogroup I (Norwalk strain), while the ONT dataset is derived from genogroup II sequences. The level of divergence between GI & GII viruses could be between 40-60% (amino acid). My concern is not necessarily the ability for Variabel to identify sub-consensus variants but rather poor or incorrect alignments where GII derived reads have been mapped to the GI reference genome.

Could the authors please provide some assurance that the results haven't been impacted by the choice of this divergent reference sequence?

We thank the reviewer for the feedback. With respect to the reviewer's concern about choosing correct reference genome, the norovirus dataset is a cross-patient dataset with norovirus GII-positive samples. The reference genome we used for read alignment of this dataset is NCBI Reference Sequence: NC_039477.1 (Norovirus GII, complete genome), which we stated in the manuscript (Page 14, Methods section, Quality control and read alignment subsection)

Although the results have not been impacted since the norovirus reference genome used for read alignment matched the norovirus in the samples, given this generated confusion, we have updated the following text in the manuscript for clarification.

> (Methods, Dataset descriptions, Page 13)

The norovirus dataset includes 39 full-length amplicon sequenced cross-patient norovirus GII-positive samples on ONT sequencing platform.

> (Methods, Quality control and read alignment, Page 14)

For the norovirus dataset, the reads were aligned to the norovirus GII reference genome (NCBI Reference Sequence: NC_039477.1)

Other changes to the manuscript

We changed multiple “cross patient” to “cross-patient” in the manuscript text, figure titles, and figure legends for consistency.

We added multiple “the” to the description of dataset in the manuscript text, figure titles, and figure legends for consistency.

We have added some acknowledgements to the manuscript.

> (Acknowledgements, Page 17)

We would like to thank Dr. Michael Nute for his constructive feedback on the project.

> (Acknowledgements, Page 17)

This work was supported in part by the Big-Data Private-Cloud Research Cyberinfrastructure MRI-award funded by NSF under grant CNS-1338099 and by Rice University's Center for Research Computing (CRC).

Response to reviewers

Reviewer 2.

- 1) We have prepared a new panel in Figure S1 (S1j), where we indicate which loop residues have not been modeled in the three structures. We have furthermore added a short statement in the Results section, as suggested by Reviewer 2. “the modeled residues have B-factors well above average (see Fig. S1, which also lists the loop residues that have not been modeled).”
- 2) We have now made a zoom onto the loop region, which shows the RMSF from the MD simulations. “Interestingly, simulations of TET1, TET2 and TET3 dimeric assemblies also show that this loop is a very dynamic structural element, suggesting that at least within the archaeal TET assemblies the loop flexibility is retained, although the PPH motif is less dynamic in TET1 and TET3 than in TET2 (Fig. S12).”

In the Discussion we added: “Crystallography (Fig. S1) and MD simulations (Fig. S12) point to motion of these loops also in other TET isoforms, although the effect appears most pronounced in TET2.”

We have kept the statements relatively short, because just as reviewer 3, we feel like adding too much would compromise the readability of the paper. All the data are clearly displayed now in Figures S1 and S12.